# Transport Variability of the Brazil Current from Observations and a Data Assimilation Model

Claudia Schmid [1] and Sudip Majumder [2]

[1]National Oceanic and Atmospheric Administration
[2]Cooperative Institute for Marine and Atmospheric Studies, University of Miami, Miami, FL, USA

*Correspondence to:* Claudia Schmid (claudia.schmid@noaa.gov)

**Abstract.**

The Brazil Current transports from observations and the Hybrid Coordinate Model (HYCOM) model are analyzed to improve our understanding of its structure and variability. A time series of the observed transport is derived from a three-dimensional field of the velocity in the South Atlantic covering the years 1993 to 2015 (hereinafter called Argo & SSH). The mean transports of the Brazil Current increases from $3.8\pm2.2$ Sv (1 Sv is $10^6 m^3 s^{-1}$) at 25$^o$S to $13.9\pm2.6$ Sv at 32$^o$S, which corresponds to a mean slope of $1.4\pm0.4$ Sv per degree. Transport estimates derived from HYCOM fields are somewhat higher ($5.2\pm2.7$ Sv and $18.7\pm7.1$ Sv at 25$^o$S and 32$^o$S, respectively) than those from Argo & SSH, but these differences are small when compared with the standard deviations. Overall, the observed latitude dependence of the transport of the Brazil Current is in agreement with the wind-driven circulation in the super gyre of the subtropical South Atlantic. A mean annual cycle with highest (lowest) transports in austral summer (winter) is found to exist at selected latitudes (24$^o$S, 35$^o$S and 38$^o$S). The significance of this signal shrinks with increasing latitude (both in Argo & SSH and HYCOM), mainly due to mesoscale and interannual variability. Both Argo & SSH, as well as HYCOM, reveal interannual variability at 24$^o$S and 35$^o$S that results in relatively large power at periods of two years or more in wavelet spectra. It is found that the interannual variability at 24$^o$S is correlated with the South Atlantic Subtropical Dipole Mode (SASD), the Southern Annular Mode (SAM) and the Niño 3.4 index. Similarly, correlations between SAM and the Brazil Current transport are also found at 35$^o$S. Further investigation of the variability reveals that the first and second mode of a coupled empirical orthogonal function of the meridional transport and the sea level pressure explain 36% and 15% of the covariance, respectively. Overall, the results indicate that SAM, SASD and El Niño Southern Oscillation have an influence on the transport of the Brazil Current.

# 1 Introduction

The circulation in the South Atlantic has been studied extensively because it is an important part of the Atlantic Meridional Overturning Circulation, which consists of a northward transport of relatively warm and fresh upper ocean water of southern origin across the equator into the northern North Atlantic and a southward transport of relatively cold and salty deep water from the North Atlantic into the South Atlantic. A summary of the circulation in the South Atlantic as well as the pathways of the flow and its role in the Atlantic Meridional Overturning Circulation has been presented by Schmid (2014) and many others (references can be found in Schmid, 2014).

Herein, the focus is on the structure and variability of the Brazil Current, which is the western boundary current of the subtropical gyre in the South Atlantic. This subtropical gyre is largely governed by the Sverdrup Equation (Pond and Pickard, 1983) and is part of the super gyre (Gordon et al., 1992; de Ruijter, 1982) which connects the subtropical circulation in the South Indian and South Atlantic Oceans. Mostly, the Brazil Current follows the shelf break quite closely, but it is impacted by mesoscale variability along its pathway that can give rise to meanders that separate it from the shelf break temporarily (e.g., Schmid et al., 1995; Biló et al., 2014; Mill et al., 2015; Lima et al., 2016). As the Brazil Current reaches the confluence with the Malvinas Current it is forced away from the shelf break and ultimately feeds into the eastward South Atlantic Current (e.g., Gordon, 1989; Garzoli, 1993; Maamaatuaiahutapu et al., 1998). Just prior to this eastward turn the southward transport increases due to the contribution from the Malvinas Current. Determining source and variability of the Malvinas Current (e.g., Vivier and Provost, 1999; Spadone and Provost, 2009) as well as what happens east of the confluence is beyond the scope of this study.

Another feature of the circulation in this region is a northward flow just east of the Brazil Current that originates near the confluence and is part of a recirculation cell that feeds back into the Brazil Current. This recirculation cell has been described earlier (e.g., Stramma, 1989) and has been called the Brazil Current Front (e.g., Peterson and Stramma, 1991) as well as the

Brazil Return Current (e.g., Boebel et al., 1997).

The transport of the Brazil Current estimated in earlier studies varies from north to south (Fig. 1). This transport is within 1 Sv to 7 Sv (1 Sv is $10^6 m^3 s^{-1}$) between 19$^o$S and 22.5$^o$S in the upper 400 to 500 m and increases to about 17 Sv at 28$^o$S as the vertical extent and strength of the Brazil Current increases. Farther south the Brazil Current transports are mostly in the range of 10 to 30 Sv. Most of the estimates from earlier studies are based on quasi-synoptic sections, while some are based on time series from moorings with current meters or Inverted Echo Sounders (IES).

Previous studies of the temporal variability were typically limited in terms of the length of the time series (e.g., Rocha et al., 2013), the number of surveys (e.g., Mata et al., 2012) or derived as a time series at one location (e.g., Goni and Wainer, 2001), In addition, studies based on hydrographic measurements had to use a level of no motion or make assumptions about the barotropic flow (e.g. by prescribing a bottom velocity). The large variations in the transports from the previous studies as well as the limited knowledge about the temporal variability of the Brazil Current motivated this study on the characteristics and variability of this current at a wide range of latitudes.

Another motivation is that, as is well known, estimates of the Atlantic Meridional Overturning Circulation transports derived from various observational products and models often reveal similar amplitudes of the variability, but can have significant differences when the means are compared. For the North Atlantic, this was shown, for example, by Msadek et al. (2014). The same is the case in the South Atlantic. An important challenge for Atlantic Meridional Overturning Circulation transport calculations is the estimation of the transport in the western boundary current (the Brazil Current in the Subtropical South Atlantic). All estimates of this transport face the challenge of deriving the contributions on and often also near the shelf break. Typically, this challenge is resolved by using climatology (e.g., Garzoli et al., 2013; Majumder et al., 2016).

In summary, this study will build on the earlier results with the focus on improving the knowledge about the mean transport of the Brazil Current and its variability. In preparation for this analysis a monthly observations-based time series of three-dimensional fields of the horizontal velocity was derived. This time series covers 23 years with a horizontal grid resolution of 0.5°. The underlying dynamics of the observed variability on seasonal to interannual time scales are studied in conjunction with several ocean indices and sea level pressure as a proxy for the wind field that is forcing the subtropical gyre.

The paper is organized as follows. Section 2 describes the data and methods. Sections 3 and 4 analyze the structure and variability of the Brazil Current transport. Section 5 summarizes the results.

## 2   Data and methodology

Three oceanic data sets are used herein to derive an absolute three-dimensional geostrophic velocity field. They are profiles of temperature and salinity, subsurface velocities from float trajectories and sea surface heights. In addition, wind fields are needed to estimate the Ekman velocity that needs to be added to the geostrophic velocity prior to studying the circulation. Where these data sets come from and how they are used is described in the following.

The temperature and salinity profiles come from an array of roughly 3000 floats that drift freely in the world ocean as part of the Argo project (the goal of 3000 active floats was reached in 2007). Details on the procedures regarding data acquisition and quality control were described by Schmid (2014). An expansion of the time period by about 1.5 years over the one available in the previous study yielded 81,627 profiles with good temperature and salinity collected in the study region (Fig. 2) during 2000-2015. Profile data are available throughout most of the study region (Fig. 2a) and this data coverage does not depend on the calendar month (not shown).

The trajectory data used for the estimation of the subsurface velocity are from Argo and WOCE floats that were active in January 26, 1989 to May 19, 2016. Details on the types of floats included in the data set can be found in Schmid (2014). As before, trajectories from floats drifting in the pressure range of 800 to 1100 dbar (930 of all floats) were used to derive velocity fields as monthly climatologies following the procedures described by Schmid (2014). As for the profiles, the coverage of the study region with high-quality velocities from the float trajectories is quite good (Fig. 2b) and the data coverage does not depend on the calendar month (not shown).

In addition, daily sea surface height (SSH) fields from AVISO are used (AVISO, France, 1996). This data set consists of delayed-time absolute dynamic topography on a $1/4^o$ grid covering the time period January 1993 to December 2015. The in situ data in conjunction with the sea surface height fields are used to derive absolute geostrophic velocities as described by Schmid (2014). The first step is to establish the relationship between the dynamic height profiles (derived from Argo profiles)

and the SSH on a regular grid. Once this relationship has been determined, gridded fields of synthetic dynamic height profiles can derived. The next step is to calculate the zonal and meridional geostrophic velocity. Finally, the monthly climatology of the subsurface velocity fields from the trajectory data is used to apply a barotropic adjustment to the geostrophic velocity fields.

5    As in Schmid (2014), wind fields from the NCEP reanalysis 2 (Kanamitsu et al., 2002) are used to derive the Ekman component of the transport. Majumder et al. (2016) found that the Ekman transport computed from different wind products has only a small impact on the transports of the AMOC in the South Atlantic (their Figure 14). The resulting velocity field will be called Argo & SSH hereinafter. The volume transports of the Brazil Current is derived from these velocity fields as a monthly time series covering the years 1993 to 2015 (see Appendix A).

Monthly velocity fields from the Hybrid Coordinate Ocean Model (HYCOM, Chassignet et al., 2003) covering the same time period as the velocity field derived for this study are obtained from the Global $1/12^o$ Reanalysis and Analysis which is available online (the downloaded fields are from GLBu0.08 experiments 19.0, 19.1, 90.9, 91.0, 91.1). This model has a Mercator-curvilinear grid with 32 levels and uses the Navy Coupled Ocean Data Assimilation (NCODA) system for assimilation. Al-

15  though HYCOM is a hybrid coordinate model where depth ('z') coordinates are used in the mixed layer and density coordinates in the lower layers, the output from the model is provided on depth coordinates. Information on the model experiments downloaded for this study can be found at https://hycom.org/dataserver/gofs-3pt0/reanalysis/ and https://hycom.org/dataserver/gofs-3pt0/analysis/.

20  Finally, the Southern Annular Mode (SAM, Marshall, 2003) index, the South Atlantic Subtropical Dipole Mode (SASD, Rodrigues et al., 2015) and the Niño 3.4 index (Trenberth, 1997) as well as the sea level pressure from Modern Era Retrospective-analysis for Research and Applications (MERRA, Rienecker et al., 2011) are used for the analysis and discussion of the dynamics. The SAM index is defined as the normalized gradient of the zonal mean sea level pressure between $40^o$S and $65^o$S. The Niño 3.4 index is valid for the region $120^o$W to $170^o$W, $5^o$S to $5^o$N. The SASD index is derived from the sea surface

temperature anomalies averaged within two regions (30-40$^o$S, 10-30$^o$W and 15-25$^o$S, 0-20$^o$W) by subtracting the estimates in the northern region from those in the southern region.

## 3 Mean characteristics of the Brazil Current transport

The mean transport for the upper 800 m, as derived from the monthly Argo & SSH time series, reveals two bands of the westward southern South Equatorial Current, which are part of the wind-driven subtropical gyre and feed into the Brazil Current at two main latitudes (near 22$^o$S and around 30$^o$S, Fig. 3a). North of about 26$^o$S the Brazil Current is represented relatively poorly

in the mean field. Between 26$^o$S and 28$^o$S it becomes more visible and it is strongly developed farther south. A comparison with the mean surface velocity field presented by Oliveira et al. (2009) reveals a lot of similarity to the transport field derived herein: in the region south of about 26$^o$S Oliveira et al.'s Figure 4 shows a well developed Brazil Current while it is relatively poorly defined in 23$^o$S to 25$^o$S where they find that the mean kinetic energy is lower than the eddy kinetic energy (Fig. 6 in Oliveira et al., 2009). The reason for this is the larger variability of the location of the Brazil Current as well as its weakness

in this area as already observed by Mata et al. (2012). A contributing factor to this is the eddy variability in this region, an example is the frequent occurrence of the so-called Vitória eddy (e.g. Schmid et al., 1995; Arruda et al., 2013).

Similar to Argo & SSH, the HYCOM model also shows a strengthening of the Brazil Current from north to south, however, this strengthening starts farther north than in Argo & SSH (Fig. 3b). The main branches with westward flow in HYCOM reach

the boundary near 22$^o$S and 28$^o$S. The latter is close to the northern edge of the southern branch with westward flow in Argo & SSH. Differences in the structure of the Brazil Current are visible when comparing HYCOM with Argo & SSH. Tendentially, the Brazil Current in the model is close to the 800 m isobath. North of 25$^o$S, the mean field from Argo & SSH has the southward flow about 2$^o$ east of the 800 m isobath. HYCOM has a corresponding band of southward flow there, in addition to a more chaotic southward flow closer to the western boundary. This is consistent with the meandering of the Brazil Current in

this region, as mentioned in section 2 based on evidence from earlier studies.

Details on the latitude dependence of the transport of the Brazil Current (which has been derived following the method described in Appendix A) are shown in Figure 4. For Argo & SSH and HYCOM the means are derived from monthly time series over the full time period of 23 years. Before going into details it has to be noted that many earlier studies used varying

layer thicknesses. North of 27$^o$S the layer thicknesses are mostly smaller than 800 m and can be as small as 400 m. In support of this latitude dependence of the vertical extent of the Brazil Current the velocity structure in the Argo & SSH fields in this region indicates that the Brazil Current frequently is not well-defined below about 400 m. This is the reason for the statistics in Table 1 which show that the mean transport in 20$^o$S to 27$^o$S in the upper 400 m is almost as large as in the 0-800 m layer.

Overall, the deeper layer (400-800 m) carries less than 19% of the transport in the upper 800 m in this latitude range (both for Argo & SSH and HYCOM). This is also in good agreement with the results of Rocha et al. (2014) as well as the dynamics governing wind-driven subtropical gyres (e.g., Luyten et al., 1983, their Figure 7). While the latter study is in the North Atlantic the method can be applied in the South Atlantic as has been done by Schmid et al. (2000), for example. Farther south the transport in the deeper layer contributes almost twice as much (36% for Argo & SSH, 32% for HYCOM in 39$^o$S to 33$^o$S,

Table 1) to the transport in the upper 800 m. Based on these characteristics the transport in the upper 400 m will be used for the analysis in the region north of 27$^o$S from here on.

When comparing the mean meridional transport of the Brazil Current from Argo & SSH (black line in Fig. 4) with historical estimates (grey symbols in Fig. 4), one can detect a tendency for higher transports in some of the synoptic surveys. This

is especially common north of 31$^o$S. Potential causes for such differences could be the inclusion or exclusion of the Ekman transport, differences of the vertical integration limits, representation of transports in the portion of the Brazil Current that is in shallow areas, and the impact of mesoscale variability. These will be discussed in the following.

The computation of the contribution of the Ekman transport to the transport of the Brazil Current reveals that the former is

very small. Its magnitude amounts to less than 5% in 97% (99%) of the cases when compared with transports of the Brazil Current that exceed 1 Sv (2 Sv). Therefore, the Ekman contribution to the transport of the Brazil Current can be considered to be insignificant for these comparisons.

As stated above, the transports from earlier studies in the region north of 27$^o$S are estimated with varying layer thicknesses that mostly exceed 400 m. Because the transports from Argo & SSH are derived for the upper 400 m the transports from the earlier studies can be higher. However, this is unlikely to be the only reason for the differences (most of them are in the range of 2 to 6 Sv) because the 400-800 m layer contributes less than 19% to the transport in the upper 800 m (see above and Table 1).

An analysis of the contribution of the transport in shallow water to the total transport of the Brazil Current reveals that this contribution is small when compared with the differences between the independent transport estimates in Figure 4 (see Appendix B). The derived estimates indicate that this contribution does not exceed 2 Sv throughout the study region. Adding up the impacts of the shallow contribution and the layer thickness for the region north of 27$^o$S results in a combined effect that remains close to 2 Sv, which is still smaller than many of the differences between the transports from quasi-synoptic surveys and Argo & SSH that exist in this region.

Individual quasi-synoptic transects indicate that there is significant mesoscale variability in this latitude range (alternating 1-2 degree wide bands of southward and northward velocity with magnitude 20 to 30 $cms^{-1}$ in XBT transects), both near 22$^o$S (Mata et al., 2012) and 25$^o$S (Garzoli et al., 2013). These meridional velocities are often twice as high as the monthly mean velocity in Argo & SSH. Therefore, one can get a roughly twice as large Brazil Current transport from individual transects for a given month and year when compared with the corresponding transport from a monthly mean velocity field. Taking an average of such quasi-synoptic transports can therefore result in a larger Brazil Current transport when compared with those from Argo & SSH. An example of the impact of that variability can be seen at 24.5$^o$S in Fig. 4 (grey dot with large error bar). Adding this effect to the other two (layer thickness and shallow water contributions) can explain most of the differences between the estimates from previous studies and Argo & SSH.

Transport estimates from individual hydrographic sections taken south of 27$^o$S mostly agree well with the means from Argo & SSH. However, a few exceptions exist, including the 51.4 Sv at 36$^o$S by Zemba (1991), which is about twice as high as the

mean from Argo & SSH. This large discrepancy is not very worrisome because the mesoscale activity at this latitude is very high due to the confluence of the Brazil Current and the Malvinas Current, which typically is found within about $3^o$ of $38^oS$. Therefore, snapshots from quasi-synoptic sections can result in significantly larger transports than monthly averages.

More straightforward is a comparison of the mean transport estimates from the XBT lines (Garzoli et al., 2013, grey dots in Fig. 4) with those from Argo & SSH, because multiple estimates from transects at a given latitude will reduce the impact of high variability. For example, at $35^oS$ the mean Brazil Current transport is 12.6±2.6 Sv from Argo & SSH (Table 2). When keeping the variability at this latitude and the difference in observation period and method in mind, this result agrees very well with the 16.3±7.3 Sv derived from the XBT lines compiled by Garzoli et al. (2013) as well as the 14±7 Sv derived by

Goni and Wainer (2001) based on a TOPEX/POSEIDON ground track crossing the Brazil Current near $35^oS$ (their Figure 7).

For the historical transport estimates the latitude dependence between $19^oS$ and $32^oS$ corresponds to a mean slope of about 1.6 Sv per degree (Fig. 1). However the characteristics in Figure 4 indicate that one can analyze the regions north and south of $25^oS$ separately. In the northern region ($20^oS$ to $25^oS$), the latitude dependence is relatively weak because the transports are

not impacted by the strong westward flow reaching the boundary in the southern region (between $25^oS$ and $32^oS$). The mean transport in the northern region from the historical studies is larger than the corresponding transport from Argo & SSH and also has a larger standard deviation (6.0±3.5 Sv versus 1.9±0.8 Sv). For Argo & SSH the largest time-averaged transport in this latitude range is 3.8±2.2 Sv at $25^oS$. In addition, the mean of 1.9±1.1 Sv at $22^oS$ from Argo & SSH is in good agreement with the mean (2.3 Sv) derived near $22^oS$ by Mata et al. (2012). Overall, the difference between the independent estimates in

the northern region is not very large when keeping the standard deviations in mind.

In the southern region the transport of the Brazil Current increases significantly from 3.8±2.2 Sv at $25^oS$ to 13.9±2.6 Sv at $32^oS$ for Argo & SSH, and from about 9 Sv to about 21 Sv for the historical estimates. For Argo & SSH and HYCOM, slopes of the transport within this latitude range are estimated by applying a linear fit for each month of the full time series. These

two sets of slopes are then used to derive their means and standard deviations. Due to the limited number of historical observations a different approach is used to derive the uncertainty of the slope. Four different estimates are derived by withholding some transport estimates from the calculation: slopes from a linear fit are calculated with and without considering transports lower than 4 Sv (such transports were measured near $25^o$S, see Fig. 1) as well as with and without transports within $0.5^o$

5    north of $25^o$S. The resulting slopes for the historical data range from 1.4 to 2.1 Sv per degree, with an average of 1.7±0.3 Sv per degree. For Argo & SSH and HYCOM the corresponding slopes are 1.4±0.4 Sv per degree and 1.9±0.9 Sv per degree, respectively. When taking the standard deviations into account, it can be concluded that the three estimates of the slope are in good agreement. This latitude-dependence is mainly due to the westward flow in the wind-driven subtropical gyre that reaches the boundary in this latitude range (Fig. 3).

In $33^o$S to $39^o$S the time-averaged transport from Argo & SSH fluctuates quite strongly around a mean of 17.3±3.5 Sv (Table 1, black line in Fig. 4). It is not likely that this is caused by changes in the southern South Equatorial Current, because most of the water transported by this current reaches the western boundary north of $33^o$S (Fig. 3). One possible cause is the Brazil Return Current (e.g., Stramma, 1989; Peterson and Stramma, 1991; Boebel et al., 1997). Other possible causes could be the

15    location of the confluence of the Brazil Current and the Malvinas Current or the mesoscale variability in the confluence region (e.g., Gordon, 1989; Garzoli, 1993; Maamaatuaiahutapu et al., 1998). The separation of the Brazil Current Front from the shelf break can be used as a proxy to track changes in the location of the confluence. For example, Goni et al. (2011) showed a time series indicating that this separation typically occurs in 34.5 to $40.5^o$S). The method for detecting the separation described in Goni et al. (2011) was used herein to determine if its location is correlated to the transport of the Brazil Current. No such

20    correlation was found (not shown). Therefore, the most likely reason for the large fluctuation is the strong mesoscale variability in this region as indicated by the high eddy kinetic energy (e.g., Oliveira et al., 2009, their figure 6). Consistent with this, both the velocity field from Argo & SSH and HYCOM have relatively high eddy kinetic energy in the region most impacted by the Brazil Malvinas Confluence (from $33^o$S on southward within about $15^o$ from the western boundary), when compared with the

boundary region north of the confluence (Supplemental Fig. 1).

The standard deviations in Figure 4 tend to increase from north to south in observation-based and model results and the highest values are found in the confluence region. Naturally, the transports from the eddy-resolving HYCOM model have larger standard deviations than those from Argo & SSH. A closer look at the variability, after removing the mesoscale signals in the time series, follows in the next section.

## 4 Temporal variability of the Brazil Current transport

In the following the full time series of the Brazil Current transports (Fig. 5) is analyzed. Three latitudes were selected for this analysis, the northernmost one is in the regime dominated by small transports and the other two are in the vicinity of the Brazil-Malvinas confluence. The main focus will be on the annual cycle (Fig. 6), which has been derived by subtracting the annual mean for each year from the individual monthly transports in that year to reduce the impact of the interannual variability. The effect of this approach is similar to a high pass filter.

### 4.1 Variability at 24°S

The transport from Argo & SSH in the upper 400 m at 24°S ranges from 0.4 Sv to 5.1 Sv with a mean of 2.3±0.9 Sv (Table 2), and reveals a relatively complicated variability, mostly with one to two transport maxima in each year (black line, Fig. 5, top). Typically, the transports are high in austral summer and low in austral winter. This can be seen more clearly in Figure 6 (black line), which shows the annual cycle represented as the anomaly of the transport. On average, the smallest transport occurs in July and the largest in March. The amplitude of the annual cycle is 0.6 Sv, with transports ranging from 1.7 Sv to 2.8 Sv (Table 3). The years for which a semiannual cycle is indicated by two transport maxima give rise to the dip of the anomaly to about 0.1 Sv in October. However, in terms of indicating the presence of a semi-annual cycle this feature does not reach the level of significance. The alternating multi-year phases with significant spectral density at semi-annual and/or annual periods is reflected in the wavelet power spectrum mainly before 2002 (Fig. 7a). Longer-periodic variability also has relatively high spectral density, primarily for periods of 2 to 4 years. Similar patterns can be seen in the wavelet spectrum for

HYCOM (Supplemental Fig. 2). Mostly, these do not reach the level of significance for both Argo & SSH and HYCOM, with the exception of a period in 1997 to 1998 in Figure 7a. Interannual variability is discussed in more detail in sections 4.4 and 4.5.

The wavelet power spectrum for SAM also reveals phases with significant energy at the semi-annual period, as well as quite persistent phases of relatively high energy at periods of one to two years (Fig. 7b). A cross wavelet analysis reveals a significant signal at the annual period in 1995-1998 (Fig. 7c) with high values of the wavelet coherence (Fig. 7d). In contrast to that, the second period (2007-2009) for which the significance level is exceeded at the annual period in Figure 7c has low values in the wavelet coherence (Fig. 7d) because the annual cycle of SAM is very weak during that time (Fig. 7b).

On average the Brazil Current transports from HYCOM are about 4 Sv larger than those from Argo & SSH, with a mean of $6.2\pm1.6$ Sv and a range of 2.7 to 10.9 Sv (Table 2). With respect to the annual cycle, Figure 6 (red line) reveals two maxima (February and September) and two minima (June and December) at this latitude. All of these are within a month of extreme values identified in the Argo & SSH record. However, the season of relatively high transport in September in HYCOM is absent in Argo & SSH (i.e., the small change bringing the transport anomaly closer to zero in the same month in Argo & SSH is not significant). In addition to that, the amplitude of the annual cycle of 0.9 Sv is 50% larger than that for Argo & SSH (Table 3). The characteristics detected in the anomalies of the transport from HYCOM are in good agreement with the wavelet spectrum for this time series which has periods of high energy at semi-annual and, to a lesser extent, at annual periods (Supplemental Fig. 2, top).

## 4.2 Variability at 35$^o$S

The meridional transports of the Brazil Current at 35$^o$S in the upper 800 m from Argo & SSH are in the range of 6.0 to 21.1 Sv with a mean of $12.6\pm2.6$ Sv (Table 2, black line in middle panel of Fig. 5). As for 24$^o$S, some years in the Argo & SSH time series have two maxima of the transport while other years have only one. Figure 6 (black line) exhibits the transport minimum in June and the maximum in December. While the amplitude of 1.2 Sv is twice as large as at 24$^o$S the standard error is about four times larger (Table 3). The standard error in Figure 6 indicates that there is no significant mean semiannual or annual

cycle at 35°S. Consistent with this, the wavelet power spectrum of the transports reveals significant powers at 3 to 9 month time scales with relatively rare phases governed by a period of 6 months and no phases with a period of 12 months that reach the level of significance (Fig. 8a). It is noted that, in 2001 to 2010, the power is relatively high at the annual period and almost reaches the level of significance around 2009. The cross wavelet analysis for Argo & SSH and SAM does not reveal a coher-

ent signal at the annual cycle (Fig. 8b, c). Phases with relatively high spectral density at periods of 2 years or more exist for Argo & SSH (Fig. 8a) as well as HYCOM (Supplemental Fig. 2), however, the power is less high than at 24°S (see section 4.1).

Time series of the Brazil Current transport derived from sea surface height anomalies by Goni and Wainer (2001) and Goni et al. (2011) also indicated that the interannual variability and mesoscale variability are very strong which makes it hard

to detect any annual cycle in observations that might exist. Goni et al. (2011) found a significant peak in a spectral analysis at the annual period. Their time series has the relative minimum (maximum) of the transport in austral winter (summer) for four of the six years (Figure 7 of Goni et al., 2011). These minima and maxima are in general agreement with those found in the Argo & SSH time series.

The HYCOM time series (red line in middle panel of Fig. 5) has larger transports and variability than the Argo & SSH time series, which yields a larger mean and standard deviation (Table 2). In addition to that, HYCOM has a significant annual cycle with an amplitude that is about three times larger than the amplitude from Argo & SSH (Fig. 6, middle). The good agreement in the timing of the maxima and minima detected in Argo & SSH as well as HYCOM indicates that a significant annual cycle might exist in the ocean but can not be resolved with observations. It is noted here, that the wavelet spectrum

from HYCOM reveals a significant signal at the annual period, mainly in 2001 to 2013 (Supplemental Fig. 2), which is similar to the time frame of an almost significant annual cycle in the wavelet analysis for Argo & SSH mentioned in the previous paragraph. A likely reason for the weak signal at the annual time scale in Argo & SSH, when compared with HYCOM, could be due to insufficient in situ observations in this region with relatively large mesoscale variability (e.g., Oliveira et al., 2009). An indication that Argo & SSH might be undersampling the variability in the Brazil Current at 35°S is that the eddy kinetic

energy in Argo & SSH is between one fifth and one quarter of the eddy kinetic energy in HYCOM (Supplemental Fig. 1). This suggests that undersampling with insitu observations could reduce the ability of Argo & SSH with respect to fully resolving the annual cycle.

### 4.3 Variability at 38°S

At 38°S, the transport in the upper 800 m from Argo & SSH cover a wider range of values than at 35°S: 6.2 to 33.4 Sv, with a mean of 20.8±4.8 Sv (Table 2; black line in bottom panel of Fig. 5). With respect to the mean annual cycle, the amplitude at 38°S for Argo & SSH is the same as at 35°S (1.2 Sv, Table 3) while the standard errors are larger (1.8 Sv versus 1.3 Sv for the monthly anomalies). While Figure 6 indicates that there is no significant mean annual or semi-annual cycle at 38°S, the wavelet power spectrum of the Brazil Current transport from Argo & SSH (Fig. 8d) reveals more prevalent phases with

significant semi-annual and annual cycles when compared with 35°S. The annual cycle at 38°S has the strongest signal in 1999-2002 and 2007-2013. The cross wavelet power spectrum (Fig. 8e) indicates that some coherence with SAM may exist at the annual period in 1995 to 2001, however, the coherence plot does not support this (Fig. 8f). At periods of 2 to 4 years phases of relatively high spectral density for Argo & SSH are more prevalent than at 35°S and smaller than at 24°S. The annual cycle from HYCOM agrees well with Argo & SSH with respect to the timing and amplitude (Fig. 6). This similarity is supported by

the wavelet analysis for HYCOM (Supplemental Fig. 2), which reveals periods with a significant annual cycle that are in good agreement with those from Argo & SSH.

    Probably, a main reason for the absence of a clear mean annual cycle at 38°S is the high variability associated with the confluence of the Brazil Current and Malvinas Current (e.g., Matano, 1993; Goni and Wainer, 2001). Similar to the situation

at 35°S, the potential for undersampling could play a role at 38°S as well. However, the eddy kinetic energy from Argo & SSH is closer to that from HYCOM (reaching between 35 and 45% of the eddy kinetic energy in HYCOM, Supplemental Fig. 1). Therefore, the issue with undersampling the mesoscale variability might be less significant at 38°S. The location of the confluence is likely to play an important role here. As mentioned in section 3, Goni et al. (2011) reported that the Brazil Current Front, which can be used to trace the confluence, was between 34.5°S and 40.5°S in 1993 to 2008. On average it was

near 38$^o$S, which is the latitude discussed here.

According to Vivier and Provost (1999) the annual migrations of the Brazil Current Front are predominantly determined by the strength of the Brazil Current, which is mainly forced by the local wind stress curl (Vivier et al., 2001). Similarly, Goni and Wainer (2001) came to the conclusion that the combination of changes of the transports of the Brazil Current and the Malvinas Current drive the migration of the Brazil Current Front and that the former has a larger influence than the latter. With respect to long-term trends of the Brazil Current Front Goni et al. (2011) suggested that transport changes of the Brazil Current and the Malvinas Current are not important for frontal migrations over the time period of about 15 years.

Spadone and Provost (2009) showed that the Malvinas Current has the highest transports in May to August near 40$^o$S. During this season, the mean annual cycle indicates that the Brazil Current has relatively small transports at 38$^o$S. The wavelet transform amplitude for the Malvinas Current near 40$^o$S presented by Spadone and Provost (2009), which overlaps with the time series presented herein, has no similarity in terms of annual or semi-annual signals with the wavelet transform amplitude derived for the Brazil Current transport at 38$^o$S. This is in agreement with the argument above that the frontal location is determined by the wind stress curl rather than the transports of these two currents.

## 4.4 Relationship to Ocean Indices

In an expansion of the analysis the interannual variability of the Brazil Current transport is studied. It is found that the differences of the transports between adjacent phases with high and low values are about 1 Sv at 24$^o$S and mostly 2 to 4 Sv at 35$^o$S (Table 2). Typically, periods of relatively low or high transports last 2 to 5 years. In addition, the transport at 24$^o$S reaches a minimum in 2000 (black line in Fig. 9a) which is followed by a maximum in 2002/2003. After a rapid drop-off followed by a period of transports close to the mean state, another transport maximum occurs in 2009/2010. When comparing 24$^o$S and 35$^o$S (black line in Fig. 9a, b), one can see periods that look similar with respect to the timing of maxima and minima of transport anomalies as well as periods without any similarity. For example, the positive anomaly in 1999/2000 and 2014 as well as the extrema in 2002/2003 and 2015 are present at both latitudes, while the two time series are very different in 2004 to 2012, for

example. The differences between 24°S and 38°S (black line in Fig. 9a, c) are even bigger, which is not very surprising because 38°S is very close to the confluence. It is noted that, mostly, HYCOM has relatively high (low) transports in the phases of high (low) transports identified in Argo & SSH.

In order to better understand what drives this variability, the relationships between various ocean indices (SAM and Niño 3.4, see section 2) and the transport at 24°S are investigated. The focus is on this latitude, because it is far enough away from the confluence. Correlations between the indices and the transport of the Brazil Current are estimated for time series filtered with different cut-off periods (Table 4). When filtering with cut-off periods of 6 months, the derived correlation coefficient between the transport and SAM is 0.5 with a lag of 5  months. For a 12-month cut-off period the correlation and lag are similar.

In agreement with this, the largest maxima of SAM (1994, 1999, 2010; red line in Fig. 9a) are followed by minima of the Brazil Current transport a few months later. Similarly, the largest minima of SAM (2002, 2013) are followed by maxima of the Brazil Current transport. The arrows in the cross wavelet spectrum (Fig. 7c) are pointing 10 to 20 degrees to the right of the downward direction in the area of relatively high power at interannual periods exceeding four years, which confirms that SAM is leading by about 5 months on interannual time scales.

      For the relationship between the transport and Niño 3.4 it is found that the lag is 6-8 months (depending on the filtering) with a correlation coefficient of 0.4. When looking at the time series, one can see that the largest El Niño events (1997/1998, 2002/2003, 2009/2010; blue line in Fig. 9a) are followed by maxima of the transport. Correspondingly, strong La Niña events (1999/2000, 2007/2008 and 2010/2011) are followed by low transports. It is too early to be sure, but it seems like the strong El

Niño of 2015/2016 could be followed by another dip in the transport of the Brazil Current (Fig. 5).

      In addition to the relationship to these remote indices, the correlation between the Brazil Current transport and SASD was derived as well. For cut-off periods of 12 months or more, the lag is very small and the correlation coefficient is 0.5 to 0.6. Because of the joint impact of SAM and El Niño, a comparison of SASD and the transport time series is difficult. Good

agreements between SASD and the transport (Fig. 9a, cyan and black lines) can be seen in the more quiescent phase with re-
spect to the remote indices (2004 to 2007) as well as during the transport maximum in 2010 that follows the 2009/2010 El Niño.

At $35^oS$, the correlations between SAM and the transport are similar to those based on transports at $24^oS$. The main differ-
ence are smaller lags. When looking at the time series (red and black line Fig. 9b), one can see multiple coinciding peaks (i.e.,
high transport when SAM is high), mainly during 1999 to 2008, which can explain the small lags. A contributing factor can
be the role the subtropical wind field, for which SAM can be seen as a proxy because $40^oS$ is used as the northern latitude to
derive this index. On the one hand, this wind field leads to the strengthening of Brazil Current on the way from $24^oS$ to $35^oS$
(Fig. 4) due to the flow in the westward southern South Equatorial Current (Fig. 3). Therefore, the variability of that zonal flow
will have an increasing impact on the variability of the Brazil Current itself as the flow strengthens on the way to the south.
On the other hand, this wind field plays a role for the location of the confluence (e.g., Wainer et al., 2000) as well as for the
contribution of the Brazil Return Current to the transport of the Brazil Current (e.g., Stramma, 1989; Peterson and Stramma,
1991; Boebel et al., 1997).

At $38^oS$ (Fig. 9c), no significant correlations between the Brazil Current transport and the indices were found, which can be
attributed to the large variability in close proximity of the confluence. In the next section, the analysis on the role that SAM and
El Niño Southern Oscillation (ENSO) play with respect to forcing the variability of the Brazil Current transport is expanded.

## 4.5 Relationship between sea level pressure and meridional transport

A coupled empirical orthogonal function (EOF) analysis of the anomaly of the sea level pressure (SLPA) in a large region,
including the Southern and tropical Atlantic and Pacific, and the anomaly of the meridional meridional transport (TVA) in
the upper 800 m in the western South Atlantic (60 to $30^oW$, 40 to $20^oS$, which includes the Brazil Current) is performed to
understand their covariability. The details on this method can be found, for example, in Bretherton et al. (1992). The use of a

bigger domain for SLP is useful to understand large scale forcing and to assess the possibility of any teleconnection pattern (Wallace et al., 1992). The coupled EOF method used herein is widely used in climate studies to identify coupled patterns between two fields.

Figure 10 shows first mode of the coupled EOF with the heterogeneous correlation maps which represent the spatial pattern for SLPA as well as the homogenous correlation maps which represent the spatial pattern for TVA (panels a, b). The normalized temporal expansion coefficients for the this mode, which explains 36% of the covariance, is shown in Figure 10c. The spatial pattern of the homogeneous correlation (Fig. 10b) has the largest correlations in the region dominated by the Brazil Current. The spatial pattern of the heterogeneous correlation (Fig. 10a) reveals a quite strong zonal symmetry throughout the

South Pacific and Atlantic, with the exception of the region south of South Africa and the tropics. South of the center of the subtropical gyres this pattern is associated with SAM, both in the Atlantic and the Pacific. In addition, the structure in the Atlantic reflects the variability in the subtropical gyre, with larger correlations in the region dominated by the Brazil-Malvinas confluence and the South Atlantic Current (near $40^{o}$S) as well as in the region where the southern South Equatorial Current is found (near $20^{o}$S). High correlations are also present in the western tropical Pacific, mainly in the Niño 3 region (within

five degrees of the equator between $170^{o}$W and $120^{o}$W), which suggests that remote teleconnections from this region (e.g., Mo and Ghil, 1986; Lopez et al., 2016) may play a role. However, it is noted that SAM is not very sensitive to the inclusion or exclusion of the Pacific region when deriving the meridional pressure gradient (see definition of SAM in section 2). This fact will make it harder to determine how important teleconnections are for this mode. More on teleconnections will follow after looking at the second mode.

The patterns of the heterogeneous and homogeneous correlation maps are robust in the sense that they do not depend much on the filtering. The main impact of varying the filtering is that the covariance explained as well as the correlation of the temporal expansion coefficients decreases with decreasing cut-off period (not shown). This is not surprising because remote signals loose their strength as they propagate over long distances and thus can be masked by regional higher-frequency variability if it

is not removed by filtering. This leads to the conclusion that remote forcing has a larger impact on long term variability than on short term variability.

The time series of the normalized temporal expansion coefficients reveal phases with high amplitude and good agreements between TVA and SLPA in 1997 to 2003 and, to a lesser extent, 2008 to 2013 (Fig. 10c). During both phases, the timing of the peaks are in good agreement, yielding an overall correlation of 0.7 for the temporal expansion coefficients. These phases coincide with relatively high amplitude of the SAM index (Fig. 10d). In periods with relatively low amplitudes of SLPA (1994 to 1996, 2004 to 2007 and 2012 to 2013) the relationship between SLPA and TVA is weak or even absent and the SAM index has a relatively small amplitude. However, there is some similarity between SAM and TVA in these periods, which is consistent with the correlations between SAM and the transport of the Brazil Current presented in section 4.4.

The impact of SAM on the transport of the Brazil Current can be understood as follows. During periods of positive SAM, the westerly winds are stronger because of a more strongly developed low pressure system centered near $50^o$S which gives rise to a relatively strong South Atlantic Current. Simultaneously, the subtropical high is stronger during the positive phase of SAM which results in easterly surface wind anomalies (Thompson and Wallace, 2000). This results in a strengthened subtropical gyre and thus a stronger western boundary current, in this case the Brazil Current.

For the second mode of the coupled EOF, which explains 15% of the covariance, the spatial pattern of the heterogeneous correlation (Fig. 11a) consists of strong zonal gradient in the tropical Pacific, with the lowest values east of the highest values. A positive anomaly is centered near the Drake Passage between Antarctica and the tip of South America. This pattern is similar to the first Pacific South American Mode (PSA1, one of the teleconnection patterns described in previous studies; e.g., Mo and Ghil, 1986; Lopez et al., 2016). This mode has been described as a response of the southern hemisphere to ENSO (Karoly, 1989) via a Rossby Wave train from the tropical Pacific to the Drake Passage.

The pattern of the homogenous correlation (Fig. 11b) is characterized by smaller spatial scales than for the first mode and the higher values near the western boundary are mainly limited to the region north of about $32^oS$. This explains why significant lagged correlations between the transport of the Brazil Current and the Niño 3.4 index were found at $24^oS$ but not at $35^oS$ or $38^oS$ (section 4.4 and Table 4).

The normalized temporal expansion coefficients for the second mode exhibit a strong correlation, with a correlation coefficient of 0.8 (Fig. 11c). The largest peak occurs in 1997/1998 and can be attributed to the very strong El Niño during that period (Fig. 11d). Another very strong El Niño event occurred in 2015/2016, which can be associated with the increase of the expansion coefficients near the end of the time period analyzed herein (an added year in the time series would be needed to fully

capture this event). Both of these El Niño events had the strongest signal in the eastern and central Pacific (in the Niño regions 3 and 3.4). During the times with moderate (1994/1995, 2002/2003 and 2009/2010) or weak (2004/2005 and 2006/2007) El Niño events the two temporal expansion coefficients also agree very well.

The changes of the SLP prior and during the 1997/1998 El Niño event are described in the following. In February to April

1997 an area of low pressure in the western tropical Pacific weakens as it expands southward into the subtropical Pacific and shifts eastward. This sets up a low anomaly of the SLP in the subtropical Pacific that also shifts eastward until August. These changes trigger meandering in the gradient zone between the subtropical high and the low pressure farther south. As these waves propagate eastward they cause changes in SLP in the subtropical South Atlantic that result in anomalous large or small gradients of the SLP which will increase or decrease the wind, respectively. These changes, in turn, will have an impact on the

circulation and therefore the transport of the Brazil Current.

Similar, but weaker, southward expansions of the low pressure in the western Pacific are also present in years without an El Niño, for example in 2000. The difference is that the the pressure remains low in the western equatorial Pacific and that the expansion has less impact on the subtropical South Pacific as well as the circumpolar region with the strong meridional SLP

gradients. As a result, eastward propagating waves do not extend as far north and the impact on the subtropical South Atlantic is smaller.

These results are in agreement with conclusions from a study by Lopez et al. (2016). They suggested that atmospheric Rossby waves originating in the tropical Pacific can travel south-eastward and reach the South Atlantic via the Drake Passage (their Figure 5). The observations presented herein, indicate that this is a likely reason for a significant part of the variability of the transport of the Brazil Current.

## 5 Summary and Conclusions

The analysis of a three-dimensional field of the horizontal velocity derived from observations covering 1993 to 2015 as well as velocity fields from HYCOM expands the knowledge of the spatial and temporal variability of the transport in the Brazil Current.

Consistent with previous studies, it is found that the mean transport of the Brazil Current as derived from Argo & SSH varies significantly with latitude, with smaller transports in the north ($1.9\pm0.8$ Sv in $20^{o}$S to $25^{o}$S), where this current originates and larger transports in the south near the confluence region ($17.3\pm3.5$ Sv in $33^{o}$S to $39^{o}$S). Between $25^{o}$S and $32^{o}$S, the transport from Argo & SSH increases gradually with a slope of $1.4\pm0.4$ Sv per degree. This increase is primarily due to westward

transports of the southern South Equatorial Current that reaches the western boundary largely within this latitude range. In principle, this is consistent with the Sverdrup balance. Farther south, the transport in Argo & SSH varies quite strongly from latitude to latitude, with an overall tendency to increase. This can be attributed to the Brazil Return Current that feeds water back into the Brazil Current as well as the Brazil-Malvinas confluence. In HYCOM, the transport increases with latitude as well. A comparison with Argo & SSH shows that the main differences are that the transports in HYCOM tend to be higher, the

increase of the transport starts farther north, and the slope between $25^{o}$S and $32^{o}$S is a bit larger ($1.9\pm0.9$ Sv per degree).

The observations reveal an annual cycle with a transport maximum in austral summer and a transport minimum in austral winter at $24^{o}$S, $35^{o}$S, and $38^{o}$S (Figs. 5 and 6). However, it is found that the significance of the mean annual cycle decreases with increasing latitude (Fig. 6). In agreement with this, a wavelet analysis indicates that phases of an annual cycle exist at

all three latitudes, but their prevalence decreases with increasing latitude (Figs. 7 and 8). Consistent with this, the time series (Fig. 5) also reveals strong interannual variability, both in terms of shifts in the annual mean and in the timing of the highest and lowest transports. Mostly, the characteristics of the temporal variability at these time scales in HYCOM are similar to those in Argo & SSH. The main difference is that HYCOM has a weak semiannual cycle at $24^{o}$S and a stronger annual cycle at $35^{o}$S

(Fig. 6).

With respect to the interannual variability it is found that the meridional transport of the Brazil Current switches from relatively high to relatively low values roughly every two to four years in the time series from Argo & SSH that were smoothed

with a one year low-pass filter (Fig. 9b). Table 2 shows statistics of such phases many of which are captured both by Argo & SSH and HYCOM. The power spectrum from the cross wavelet transform at $24^{o}$S and $35^{o}$S for Argo & SSH (Fig. 7c and 8c) and HYCOM (Supplemental Fig. 2) show weak signs for the presence of such variability that mostly do not quite reach the level of significance.

Time series smoothed with a filter using a 6 to 12 month cut-off period reveal correlations of the Brazil Current transport with SAM that are within the 95% confidence interval with lags of about 6 months at $24^{o}$S (section 4.4, Table 4). For the Niño 3.4 index the correlations with the transport remain significant while being slightly smaller with a larger lag of 8 months.

The first and second mode of the coupled EOF between the meridional transport in the Brazil Current region and the sea

level pressure provides insight with respect to the atmospheric forcing. The first mode (Fig. 10) explains 36% of the variance and supports the influence of the tropical Pacific on SAM while the second mode, which explains 15% of the variance indicates that ENSO has an impact on the meridional transport (Fig. 11), especially during strong events like the 1997/1998 El Niño.

*Data availability.* The Argo & SSH velocity fields and transport estimates for this study will be made available online via http://www.aoml.noaa.gov/phod/argo/argo_and_science.php. Until that is accomplished, the data will be made available upon request to the corresponding author.

*Competing interests.* none

*Acknowledgements.* This paper was in part funded by the Climate Observation Division, Climate Program Office, National Oceanic and Atmospheric Administration (NOAA), U.S. Department of Commerce and the Atlantic Oceanographic and Meteorological Laboratory of the National Oceanic and Atmospheric Administration. This research was also carried out in part under the auspices of the Cooperative Institute for Marine and Atmospheric Studies (CIMAS), a cooperative institute of the University of Miami and the National Oceanic and Atmospheric Administration, cooperative agreement NA10OAR432013. The authors would like to thank the researchers and technicians involved in the Argo project for their contributions to generating a high-quality global sub-surface data set.

Argo data were obtained from a Global Data Assembly Centre (Argo GDAC, doi: 10.17882/42182). Altimeter products were produced by Ssalto/Duacs and distributed by AVISO, with support from Cnes (www.aviso.altimetry.fr/ducas/). Publicly available output from a global high-resolution HYCOM run with data assimilation (see hycom.org and section 2) is used. Funding for the development of HYCOM has been provided by the National Ocean Partnership Program and the Office of Naval Research. Data assimilative products using HYCOM are funded by the U.S. Navy. Computer time was made available by the DOD High performance Computing Modernization Program. The velocity product derived from surface drifters is available at http://www.aoml.noaa.gov/phod/dac/dac_meanvel.php. The Southern Annular Mode (SAM) index was obtained from http://www.nerc-bas.ac.uk//icd//gjma//sam.html. The El Niño index was obtained from NOAA and can be found at http://www.esrl.noaa.gov/psd/data/climateindices/list/. The MERRA products are available at http://disc.sci.gsfc.nasa.gov/mdisc/data-holdings/merra/merra_products_nonjs.shtml.

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

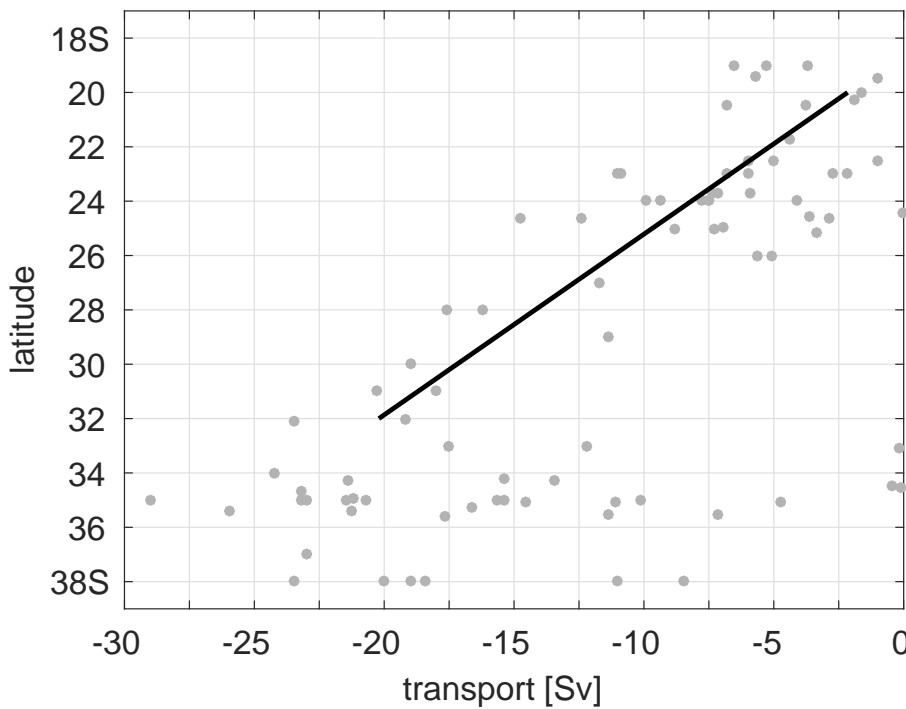

**Figure 1.** Previously published estimates of the Brazil Current transports as a function of latitude. The line with a slope of about 1.6 Sv per degree is a fit to the transports measured in $19^oS$ to $32^oS$. The sources of the transport estimates are: Fisher (1964), Signorini (1978), Miranda and Castro Filho (1979), Miranda and Castro Filho (1981), Evans et al. (1983), Evans and Signorini (1985), Gordon and Greengrove (1986), Garzoli and Garraffo (1989), Gordon (1989), Stramma (1989), Garfield (1990), Peterson (1990), Stramma et al. (1990), Zemba (1991), Garzoli (1993), Campos et al. (1995), Maamaatuaiahutapu et al. (1998), Müller et al. (1998), Jullion et al. (2006), Mata et al. (2012), Garzoli et al. (2013) and Biló et al. (2014).

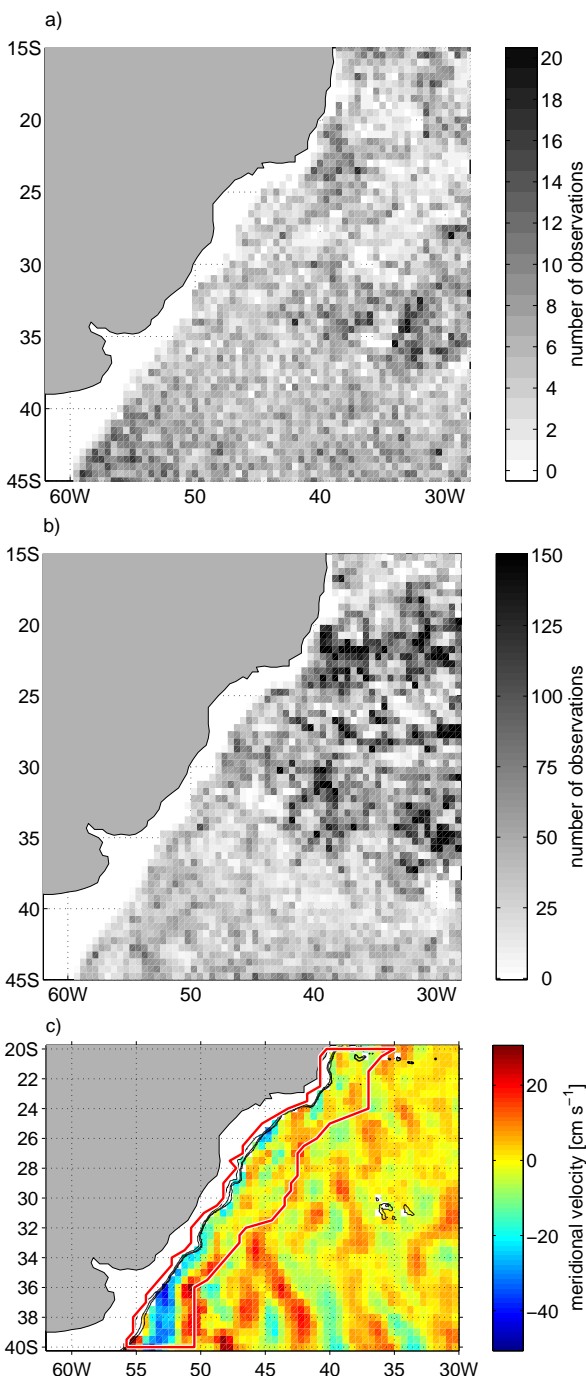

**Figure 2.** a) Availability of Argo profiles with temperature and salinity in the study region for observations collected in the years 2000 to 2015. b) Availability of trajectory observations in the study region for observations collected during January 1992 to April 2016. c) Meridional velocity in the surface layer from Argo & SSH for January 2015. The coastline as well as the 400, 800 and 1000 m isobaths are shown. The region encompassed by the red line indicates the search area for the southward flow of the Brazil Current. The bin sizes are $0.5^o$ by $0.5^o$.

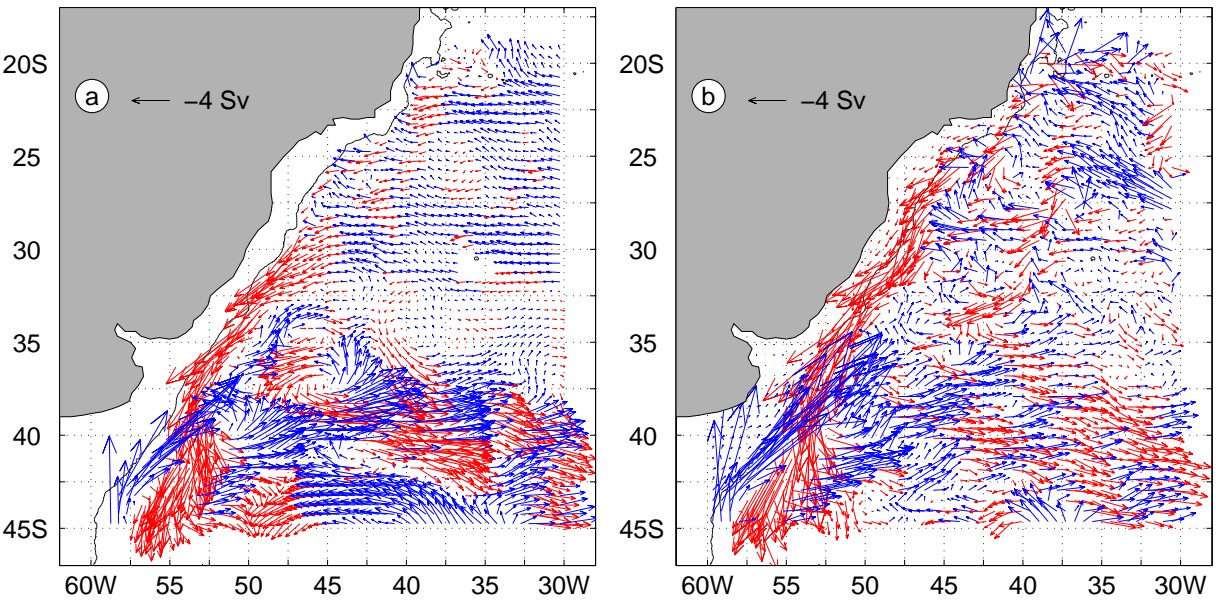

**Figure 3.** Climatological transport in the upper 800 m of the southwestern South Atlantic based on Argo & SSH (a), HYCOM (b). Red (blue) vectors indicate southward (northward) meridional transports. The 800 m bathymetry contour is also shown. It has to be noted that for HYCOM the resolution of $1/12^o$ has been reduced to match the resolution of Argo & SSH ($0.5^o$) for the sake of visibility and comparability of the vectors.

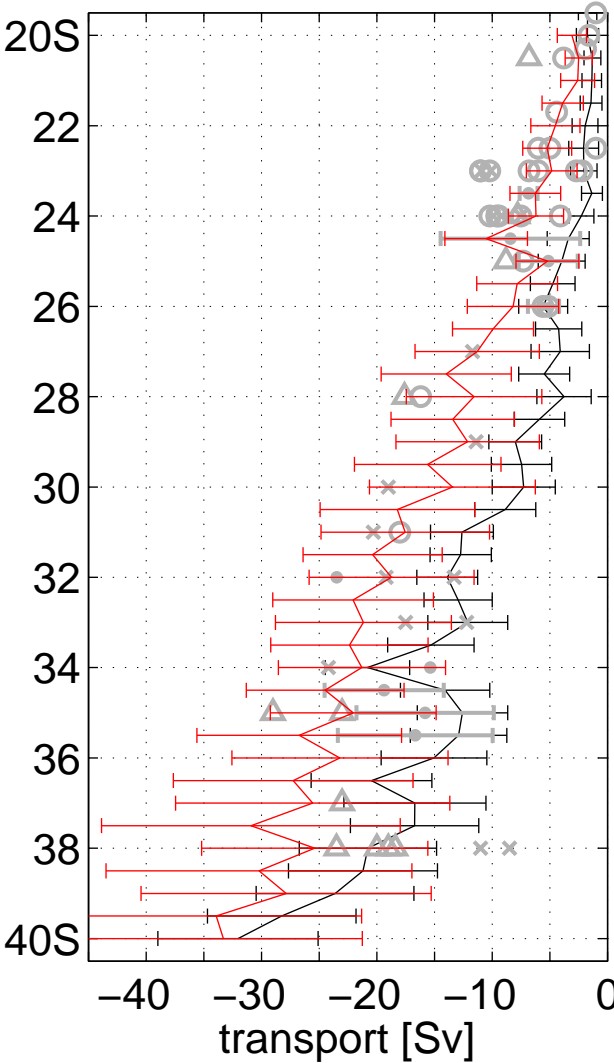

**Figure 4.** Climatological mean of the meridional transports of the Brazil Current as a function of latitude from observations (black, grey) and HYCOM (red). The black line with error bars shows the mean from Argo & SSH for a layer thickness of 400 m north of 27°S and 800 m elsewhere. Grey symbols with or without error bars are from previous studies (see Figure 1 for references). The symbols indicate if the integration depth is less than 800 m (circles), 800 m (crosses and dots) or greater than 800 m (triangles). Grey error bars are shown if the estimate is from several transects or a time series. Grey dots are based on velocity transects derived by Garzoli et al. (2013) for the purpose of estimating the Meridional Overturning Circulation transports in the South Atlantic. The red line represents the mean with error bars as derived from a combination of the HYCOM reanalysis (1993-2012) and the HYCOM analysis (2013-2015). All error bars indicate the standard deviation associated with the mean.

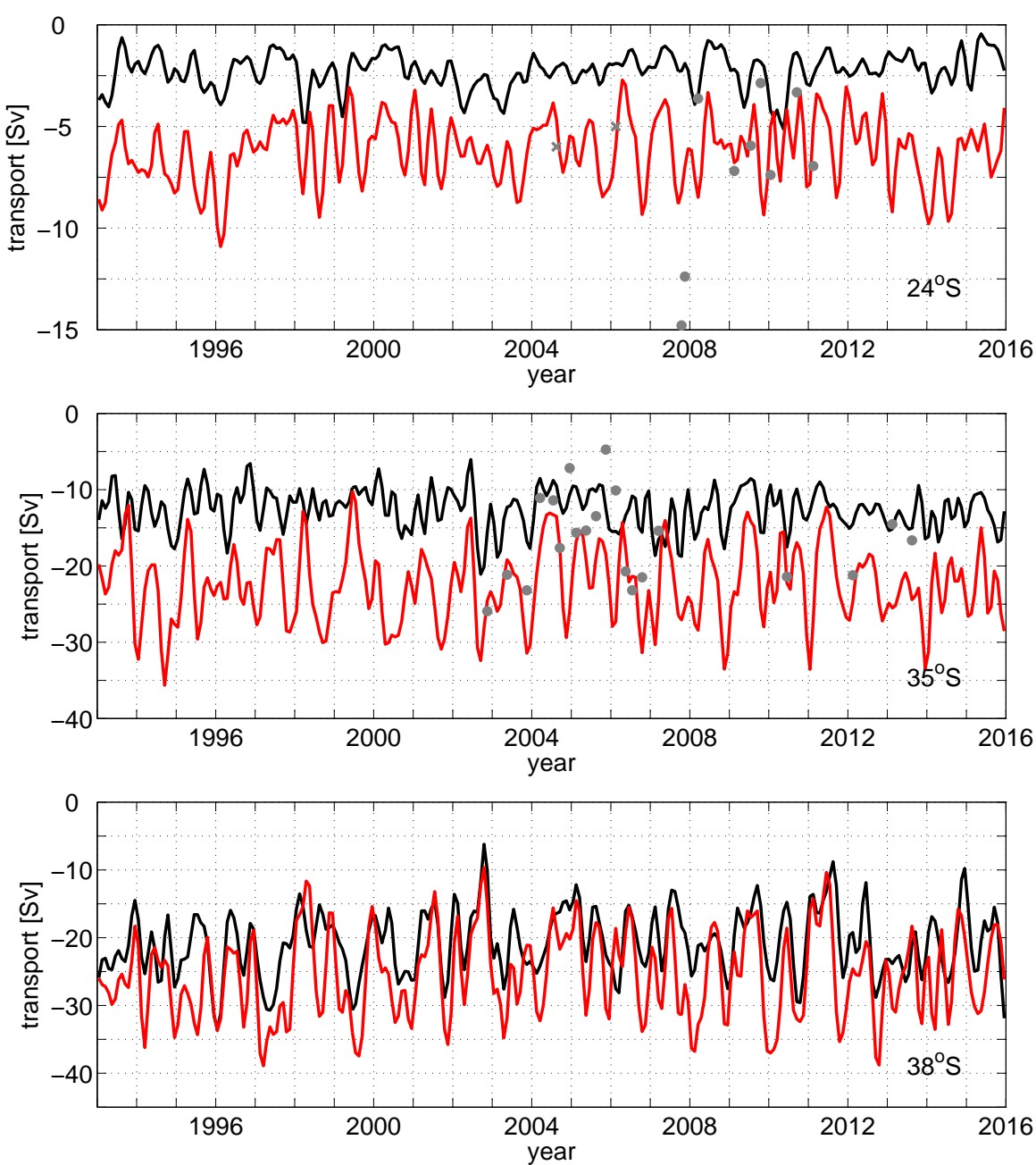

**Figure 5.** Time series of the meridional transports in the Brazil Current at $24^{o}$S, $35^{o}$S and $38^{o}$S from Argo & SSH (black) and HYCOM (red). The depth range is 0 to 400 m at $24^{o}$S and 0 to 800 m at the other latitudes. The time series were smoothed with a second order Butterworth filter (2 month low pass). Grey dots are based on transport estimates by Garzoli et al. (2013). Grey crosses indicate estimates from other studies (see Figure 1 for references).

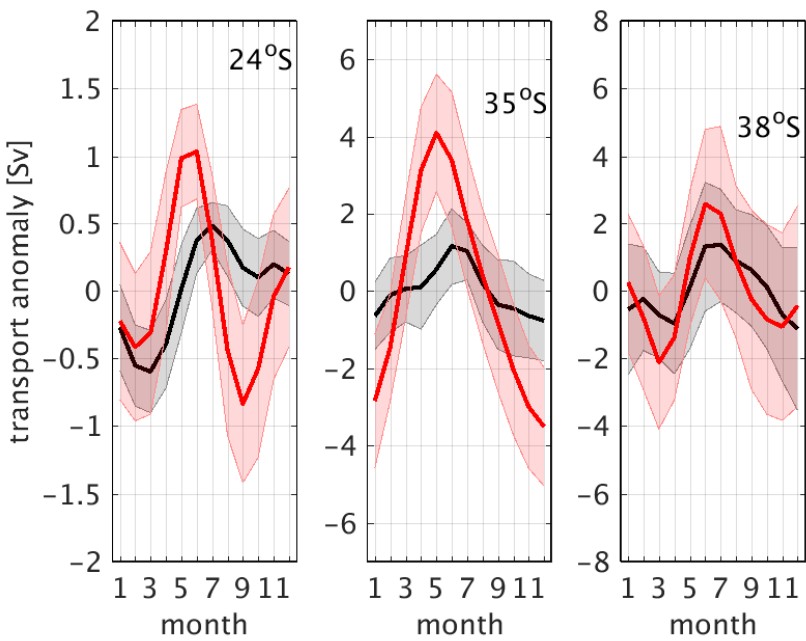

**Figure 6.** Annual cycle of the anomaly of the meridional transports in the Brazil Current derived from the time series in Figure 5 for $24^o$S, $35^o$S and $38^o$S from Argo & SSH (black) and HYCOM (red). Shading indicates standard errors.

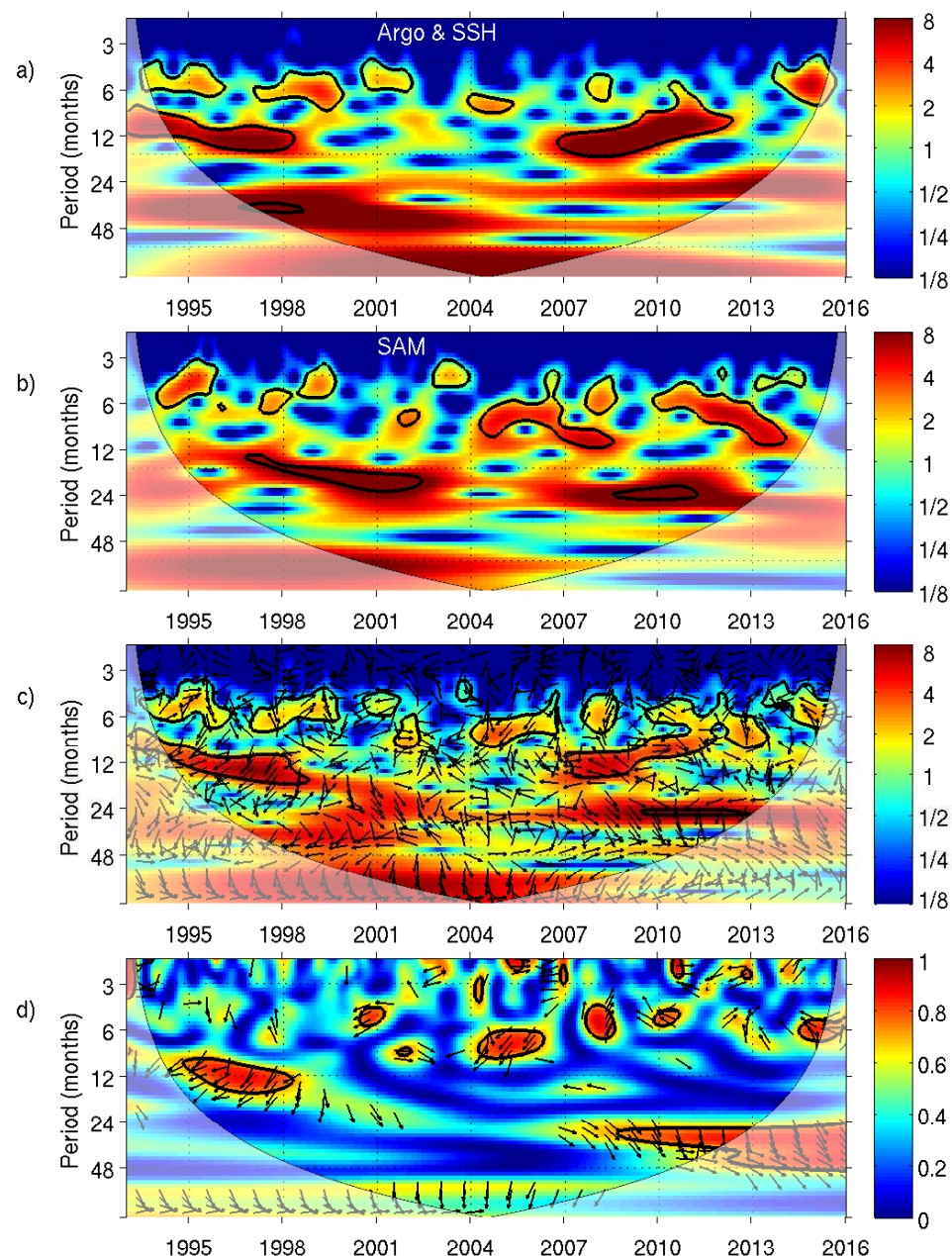

**Figure 7.** Wavelet power spectrum at $24^\circ$S for Brazil Current transport from Argo & SSH (a) and SAM (b). (c) shows the cross wavelet power spectrum between the Brazil Current transport from Argo & SSH and SAM while (d) shows the coherence. The vectors in the lower panel indicate the phase difference between them. The thick black line is the 5% significance level using the red noise model, and the thin black line indicates the cone of influence. The time series were smoothed in the same way as the time series of the Brazil Current transport in Figure 5.

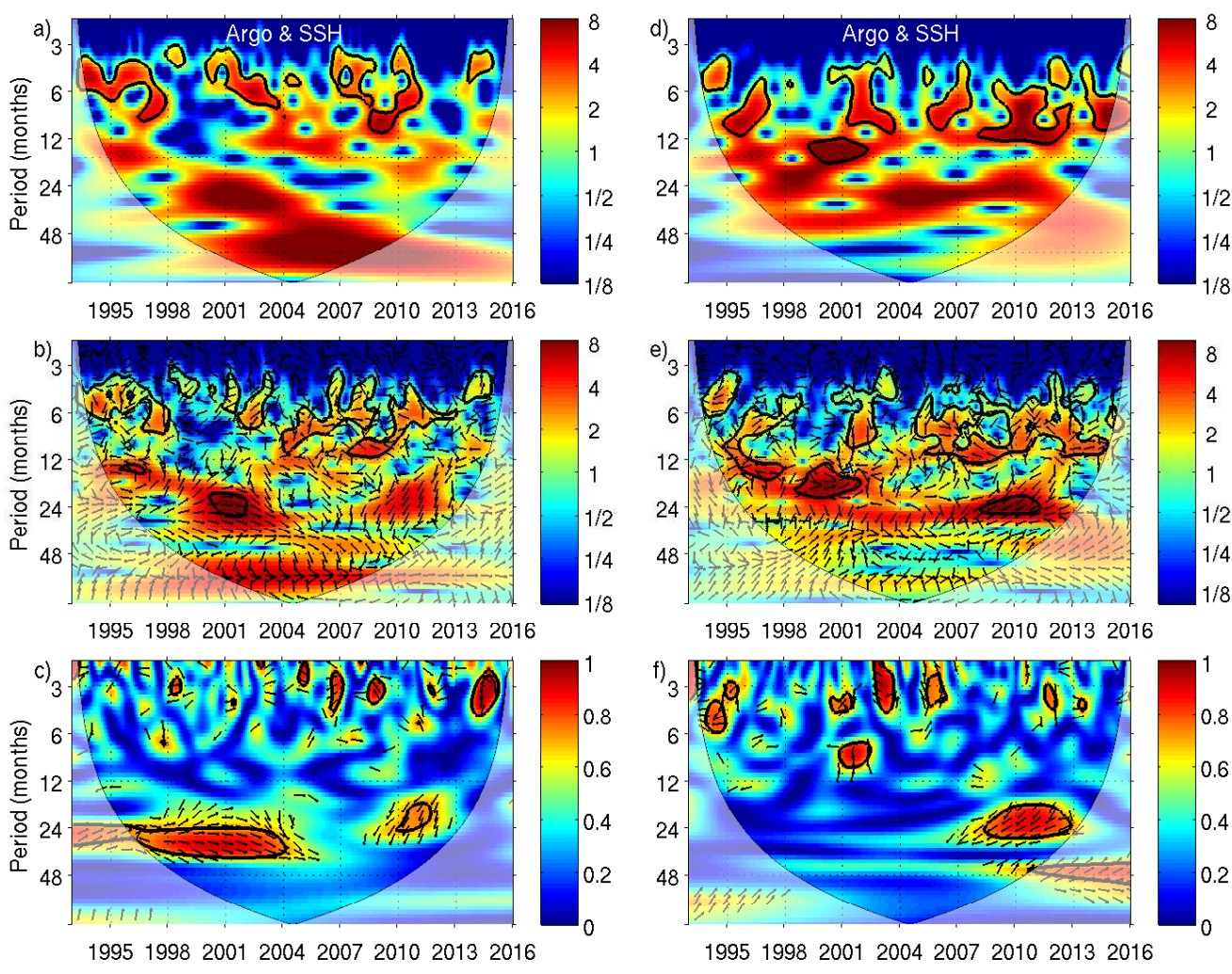

**Figure 8.** Wavelet power spectrum at $35^o$S (a, b, c) and $38^o$S (d, e, f) for Brazil Current transport from Argo & SSH (a, d). (b) and (e) show the cross wavelet power spectrum between SAM (Fig. 7b) and the Brazil Current transport from Argo & SSH for these two latitudes. (c) and (f) show corresponding the coherences. The vectors in the lower panel indicate the phase difference between them. The thick black line is the 5% significance level using the red noise model, and the thin black line indicates the cone of influence. The time series were smoothed in the same way as the time series of the Brazil Current transport in Figure 5.

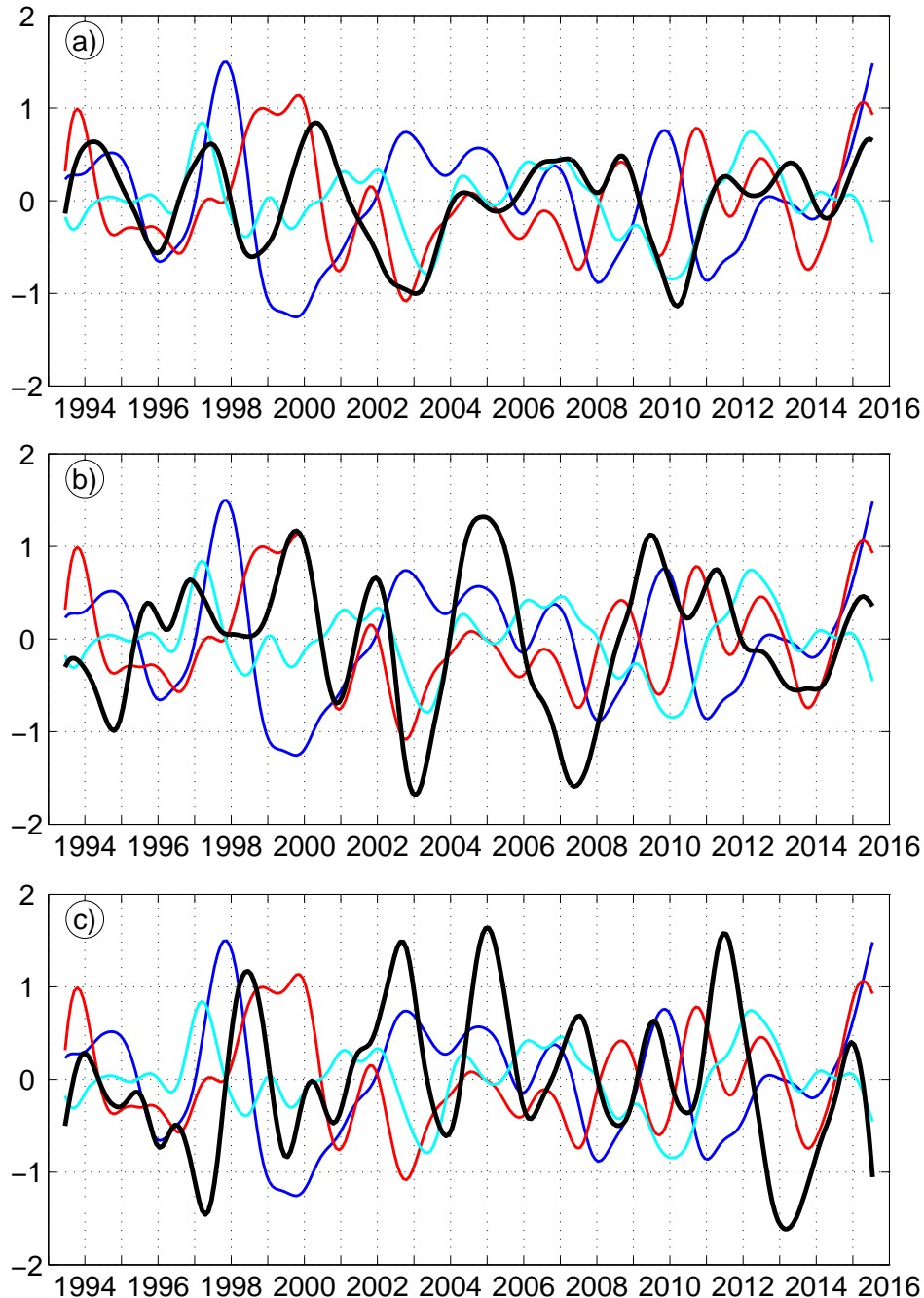

**Figure 9.** Southern Annular Mode (red) and Niño 3.4 index (blue) and South Atlantic Subtropical Dipole Mode (cyan) in comparison with meridional transports in the Brazil Current (black) derived from the time series in Figure 5. Positive (negative) anomalies for the Brazil Current transport represent low (high) transports of this current. (a) 24$^o$S, (b) 35$^o$S (the Brazil Current transport has been divided by a factor of 1.5 for easier comparison), (c) 38$^o$S (the Brazil Current transport has been divided by a factor of 2.5 for easier comparison). The linear trend has been removed from all time series and they were smoothed with a 12 month Butterworth filter.

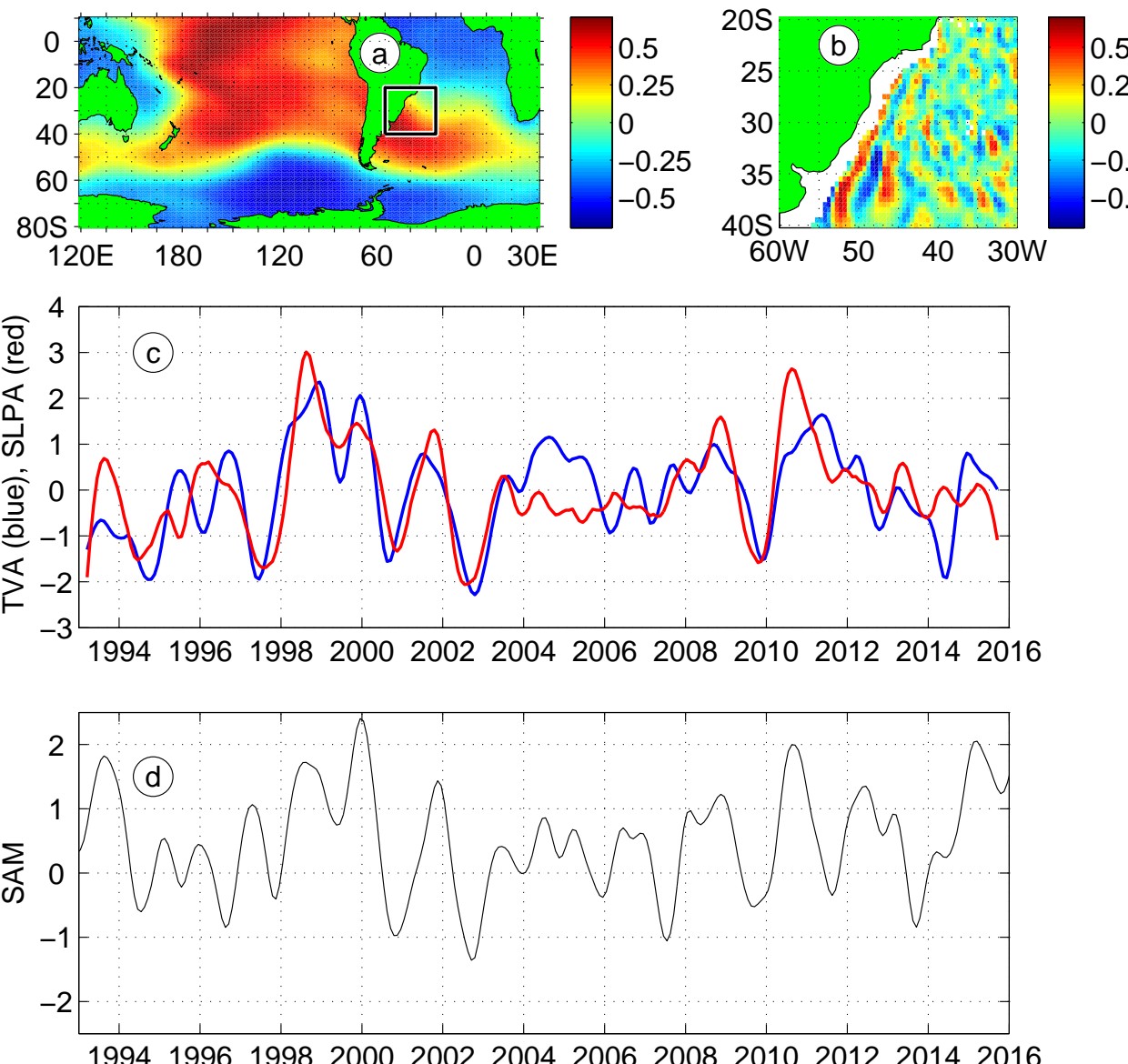

**Figure 10.** First mode of coupled EOF of the anomaly of the meridional transport (TVA) from Argo & SSH (in the box centered at $30^o$S, $45^o$W) and the anomaly of the sea level pressure (SLPA) from MERRA. The time series were filtered using a six month cut-off period and the mean annual cycle was subtracted. The spatial patterns of the heterogeneous correlation maps are presented in (a). The homogenous correlation (b), the normalized time series of the expansion coefficients (c) and the SAM index (d) are shown as well. The correlation between the expansion coefficients is 0.7, which is significant with respect to the 95% confidence level.

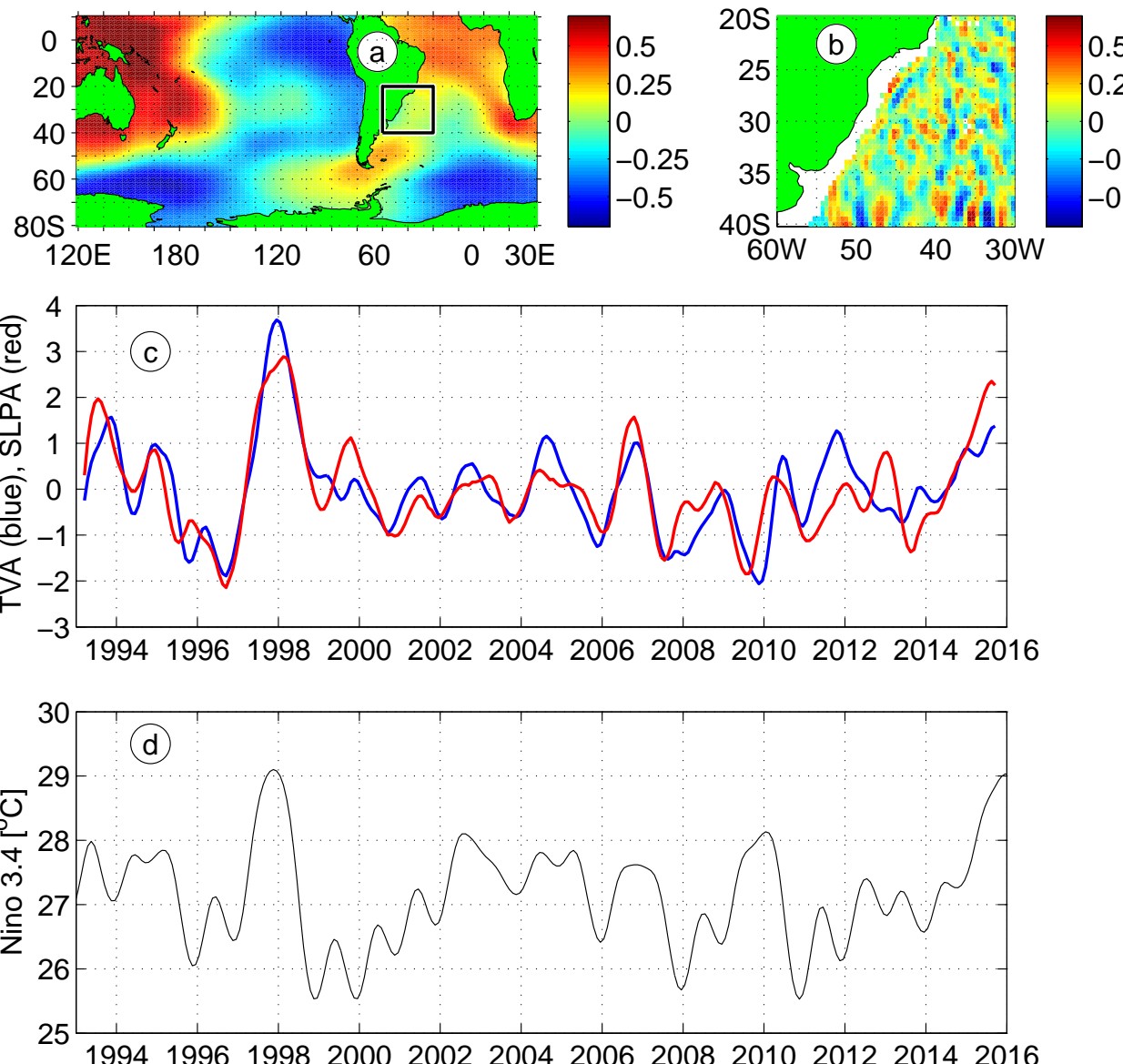

**Figure 11.** Second mode of coupled EOF of the anomaly of the meridional transport (TVA) from Argo & SSH and the anomaly of the sea level pressure (SLPA) from MERRA. The spatial patterns of the heterogeneous correlation maps are presented in (a). The homogenous correlation (b), the normalized time series of the expansion coefficients (c) and the Niño 3.4 time series (d) are shown as well. The correlation between the expansion coefficients is 0.8, which is significant with respect to the 95% confidence level. See Figure 10 for additional information.

**Table 1.** Statistics of transports in the Brazil Current region from Argo & SSH and HYCOM in various layers for two latitude ranges.

Argo & SSH

| latitude range | layer | minimum | maximum | mean | standard deviation |
|---|---|---|---|---|---|
| | [m] | [Sv] | [Sv] | [Sv] | [Sv] |
| 20 - 27$^o$S | 0-400 | 1.3 | 5.6 | 2.7 | 1.4 |
| 20 - 27$^o$S | 0-800 | 1.5 | 7.0 | 3.3 | 1.8 |
| 33 - 39$^o$S | 0-400 | 7.2 | 15.3 | 11.1 | 2.5 |
| 33 - 39$^o$S | 0-800 | 12.1 | 23.3 | 17.3 | 3.5 |

HYCOM

| latitude range | layer | minimum | maximum | mean | standard deviation |
|---|---|---|---|---|---|
| | [m] | [Sv] | [Sv] | [Sv] | [Sv] |
| 20 - 27$^o$S | 0-400 | 2.5 | 10.5 | 6.0 | 2.6 |
| 20 - 27$^o$S | 0-800 | 3.1 | 13.1 | 7.2 | 3.2 |
| 33 - 39$^o$S | 0-400 | 14.9 | 20.2 | 17.2 | 1.6 |
| 33 - 39$^o$S | 0-800 | 21.2 | 30.9 | 25.3 | 3.2 |

**Table 2.** Statistics of transports of the Brazil Current from Argo & SSH and HYCOM for the whole time series as well as for periods of relatively low or relatively high transports. Estimates are derived from the time series in Figure 5. L (H) the date column indicates low (high) transport while N indicates that a transport is not clearly high or low.

| period | latitude | Argo & SSH | | | | | |
| | | median | mean | minimum | maximum | standard deviation | standard error |
| | | [Sv] | [Sv] | [Sv] | [Sv] | [Sv] | [Sv] |
| --- | --- | --- | --- | --- | --- | --- | --- |
| 01/1993-12/2015 | 24°S | 2.2 | 2.3 | 0.4 | 5.1 | 0.9 | 0.1 |
| 01/1993-12/2015 | 35°S | 12.3 | 12.6 | 6.0 | 21.1 | 2.6 | 0.3 |
| 01/1993-12/2015 | 38°S | 20.9 | 20.8 | 6.2 | 33.4 | 4.8 | 0.6 |
| L; 07/1993-09/1994 | 24°S | 1.8 | 1.6 | 0.6 | 2.4 | 0.6 | 0.3 |
| L; 06/1996-01/1998 | 24°S | 1.7 | 1.8 | 1.0 | 2.9 | 0.5 | 0.2 |
| H; 02/1998-04/1999 | 24°S | 3.1 | 3.2 | 1.9 | 4.8 | 1.0 | 0.5 |
| L; 05/1999-02/2001 | 24°S | 1.7 | 1.7 | 1.0 | 2.7 | 0.5 | 0.2 |
| H; 03/2001-11/2003 | 24°S | 2.8 | 3.0 | 1.8 | 4.4 | 0.7 | 0.2 |
| L; 11/2005-02/2009 | 24°S | 1.9 | 1.9 | 0.8 | 3.9 | 0.7 | 0.2 |
| H; 03/2009-06/2010 | 24°S | 3.4 | 3.3 | 1.7 | 5.1 | 1.2 | 0.6 |
| L; 03/2011-10/2013 | 24°S | 1.9 | 1.9 | 1.1 | 2.5 | 0.4 | 0.2 |
| H; 01/1994-02/1995 | 35°S | 13.6 | 13.7 | 9.4 | 17.8 | 2.5 | 1.5 |
| L; 03/1995-03/2000 | 35°S | 11.4 | 11.4 | 6.6 | 18.3 | 2.3 | 0.6 |
| H; 08/2002-01/2004 | 35°S | 14.2 | 15.2 | 11.2 | 21.1 | 3.1 | 1.6 |
| L; 02/2004-11/2005 | 35°S | 10.0 | 10.5 | 8.5 | 13.1 | 1.3 | 0.6 |
| H; 12/2005-06/2008 | 35°S | 14.9 | 14.3 | 10.0 | 18.7 | 2.5 | 0.9 |
| L; 07/2008-09/2011 | 35°S | 11.5 | 11.9 | 8.5 | 17.8 | 2.5 | 0.8 |
| H; 10/2011-09/2014 | 35°S | 13.6 | 13.6 | 9.4 | 16.9 | 1.6 | 0.6 |
| H; 02/1994-11/1997 | 38°S | 22.8 | 23.1 | 16.0 | 33.4 | 4.6 | 1.4 |
| L; 12/1997-01/1999 | 38°S | 18.4 | 17.9 | 13.5 | 21.4 | 2.3 | 1.3 |
| H; 03/1999-11/2001 | 38°S | 21.7 | 22.1 | 14.7 | 30.7 | 4.7 | 1.7 |
| L; 12/2001-12/2002 | 38°S | 16.3 | 16.4 | 6.2 | 24.7 | 5.2 | 3.1 |
| H; 01/2003-05/2004 | 38°S | 23.3 | 22.5 | 15.8 | 27.6 | 2.9 | 1.5 |
| L; 06/2004-10/2005 | 38°S | 16.2 | 16.9 | 12.2 | 23.5 | 2.9 | 1.5 |
| H; 01/2007-10/2009 | 38°S | 19.5 | 19.4 | 12.3 | 27.5 | 4.0 | 1.4 |
| L; 01/2011-07/2012 | 38°S | 16.3 | 16.7 | 8.8 | 25.9 | 4.7 | 2.3 |
| H; 08/2012-11/2013 | 38°S | 24.1 | 24.3 | 19.1 | 28.8 | 2.5 | 1.3 |

|  | | HYCOM | | | | | |
| period | latitude | median | mean | minimum | maximum | standard deviation | standard error |
|  |  | [Sv] | [Sv] | [Sv] | [Sv] | [Sv] | [Sv] |
| 01/1993-12/2015 | 24°S | 6.1 | 6.2 | 2.7 | 10.9 | 1.6 | 0.2 |
| 01/1993-12/2015 | 35°S | 22.5 | 22.5 | 10.2 | 35.6 | 5.0 | 0.6 |
| 01/1993-12/2015 | 38°S | 25.4 | 25.5 | 9.6 | 38.9 | 6.4 | 0.8 |
| H; 07/1993-09/1994 | 24°S | 6.8 | 6.4 | 4.7 | 7.5 | 1.0 | 0.5 |
| L; 06/1996-01/1998 | 24°S | 5.3 | 5.4 | 4.2 | 7.0 | 0.8 | 0.4 |
| H; 02/1998-04/1999 | 24°S | 6.6 | 6.4 | 4.0 | 9.5 | 1.8 | 1.0 |
| L; 05/1999-02/2001 | 24°S | 5.1 | 5.3 | 3.1 | 8.2 | 1.4 | 0.6 |
| H; 03/2001-11/2003 | 24°S | 6.5 | 6.6 | 4.1 | 8.7 | 1.1 | 0.4 |
| N; 11/2005-02/2009 | 24°S | 5.9 | 6.0 | 2.7 | 9.3 | 1.7 | 0.6 |
| N; 03/2009-06/2010 | 24°S | 5.7 | 6.1 | 3.9 | 9.4 | 1.5 | 0.8 |
| L; 03/2011-10/2013 | 24°S | 5.5 | 5.6 | 3.1 | 9.2 | 1.6 | 0.6 |
| H; 01/1994-02/1995 | 35°S | 27.3 | 26.6 | 19.8 | 35.6 | 5.0 | 2.9 |
| L; 03/1995-03/2000 | 35°S | 22.5 | 22.0 | 10.2 | 30.0 | 4.7 | 1.2 |
| H; 08/2002-01/2004 | 35°S | 24.8 | 25.3 | 19.2 | 32.4 | 4.1 | 2.0 |
| L; 02/2004-11/2005 | 35°S | 17.5 | 18.6 | 13.1 | 29.4 | 4.5 | 2.0 |
| H; 12/2005-06/2008 | 35°S | 24.1 | 23.1 | 14.0 | 31.4 | 4.8 | 1.8 |
| L; 07/2008-09/2011 | 35°S | 20.8 | 21.0 | 12.3 | 33.6 | 5.7 | 1.9 |
| H; 10/2011-09/2014 | 35°S | 23.7 | 23.5 | 18.3 | 33.5 | 3.5 | 1.2 |
| H; 02/1994-11/1997 | 38°S | 29.7 | 28.7 | 18.9 | 38.9 | 5.3 | 1.6 |
| L; 12/1997-01/1999 | 38°S | 18.8 | 20.7 | 11.6 | 31.6 | 6.5 | 3.8 |
| H; 03/1999-11/2001 | 38°S | 27.8 | 26.8 | 13.2 | 37.5 | 6.5 | 2.3 |
| L; 12/2001-12/2002 | 38°S | 19.5 | 20.3 | 9.6 | 31.3 | 6.5 | 3.9 |
| H; 01/2003-05/2004 | 38°S | 28.0 | 27.5 | 21.1 | 34.8 | 4.0 | 2.1 |
| L; 06/2004-10/2005 | 38°S | 19.4 | 20.4 | 14.5 | 31.4 | 4.3 | 2.2 |
| H; 01/2007-10/2009 | 38°S | 24.9 | 24.8 | 15.7 | 36.8 | 6.3 | 2.2 |
| L; 01/2011-07/2012 | 38°S | 21.5 | 22.2 | 10.4 | 35.3 | 7.3 | 3.5 |
| H; 08/2012-11/2013 | 38°S | 28.2 | 27.9 | 18.3 | 38.8 | 5.7 | 3.0 |

**Table 3.** Statistics and characteristics of the annual cycle of transports of the Brazil Current. Estimates are derived from the time series in Figure 5 (for Argo & SSH, see text and Fig. 6).

| based on | amplitude [Sv] | standard error [Sv] | minimum [Sv] | maximum [Sv] |
|---|---|---|---|---|
| 24$^o$S, 0-400 m, mean | | | | |
| Argo & SSH | 0.6 | 0.3 | 1.7 | 2.8 |
| HYCOM | 0.9 | 0.6 | 5.2 | 7.0 |
| 24$^o$S, 0-400 m, anomaly | | | | |
| Argo & SSH | 0.6 | 0.3 | -0.6 | 0.5 |
| HYCOM | 0.9 | 0.6 | -0.8 | 1.0 |
| 35$^o$S, 0-800 m, mean | | | | |
| Argo & SSH | 1.2 | 1.4 | 15.1 | 17.6 |
| HYCOM | 3.8 | 1.8 | 18.4 | 26.0 |
| 35$^o$S, 0-800 m, anomaly | | | | |
| Argo & SSH | 1.2 | 1.3 | -1.1 | 1.4 |
| HYCOM | 3.8 | 1.7 | -3.5 | 4.1 |
| 38$^o$S, 0-800 m, mean | | | | |
| Argo & SSH | 1.2 | 2.2 | 19.4 | 21.9 |
| HYCOM | 2.4 | 2.7 | 22.9 | 27.6 |
| 38$^o$S, 0-800 m, anomaly | | | | |
| Argo & SSH | 1.2 | 1.8 | -1.2 | 1.3 |
| HYCOM | 2.4 | 2.2 | -2.1 | 2.6 |

**Table 4.** Correlations between various indices and the transport of Brazil Current (BCT). Time series of the Brazil Current and the indices for the 12 month filter are shown in Figure 9. SASD = South Atlantic Subtropical Dipole Mode; Sam = Southern Annular Mode. Only significant correlations are shown. CL = confidence limit.

| filter | correlation | lag | 95% CL |
|---|---|---|---|
| BCT at 24$^o$S and SAM | | | |
| 6 month | 0.5 | 5 | 0.2 |
| 12 month | 0.4 | 6 | 0.2 |
| BCT at 24$^o$S and SASD | | | |
| 6 month | 0.4 | 5 | 0.1 |
| 12 month | 0.5 | 1 | 0.2 |
| 18 month | 0.6 | 0 | 0.4 |
| 24 month | 0.6 | 2 | 0.2 |
| BCT at 24$^o$S and Niño 3.4 index | | | |
| 6 month | 0.4 | 8 | 0.2 |
| 12 month | 0.4 | 8 | 0.3 |
| 18 month | 0.4 | 6 | 0.2 |
| BCT at 35$^o$S and SAM | | | |
| 6 month | 0.4 | 3 | 0.3 |
| 12 month | 0.5 | 2 | 0.2 |
| 18 month | 0.5 | 1 | 0.1 |
| 24 month | 0.5 | 0 | 0.3 |

## Appendix A:  Details on how the Brazil Current transport is estimated

Transport profiles in grid boxes that have a water depth of less than 1000 m in their center are excluded. This means that at most latitudes, the Argo & SSH data set has a profile of the transport within $0.25^o$ of the 600 m isobath. The search area for the Brazil Current is indicated by the red line in Figure 2c that encompasses the region near the shelf break where this current is typically found. It extends east of the climatological mean core of the Brazil Current to allow for its meandering. The procedure is to pick the westernmost southward current for estimating the transport unless it is not part of the continuous southward flow. The latter situation is mostly encountered in the northern part of the domain, where a single grid box with southward velocity might exist at the shelf break while the boxes south and north of it do not support treating this box as part of the Brazil Current. An example of a situation like this near $20^o$S was studied by Schmid et al. (1995). Many others also looked at the zonal position of this current (some recent studies on this topic are Biló et al., 2014; Mill et al., 2015; Lima et al., 2016). The Brazil Current transports are derived by integrating the meridional velocity within the identified longitude range at each latitude.

## Appendix B:  Quantifying uncertainties of the Brazil Current transport

Previous studies showed that the velocity field from Argo & SSH reproduces the features of the circulation in the South Atlantic (Schmid, 2014) and can be used to derive the integrated transports associated with the Meridional Overturning Circulation at multiple latitudes (Majumder et al., 2016). As Argo & SSH is used herein to study the variability of the transport in the Brazil Current it is important to know what uncertainties exist. Quasi-synoptic XBT transects as well as output from the HYCOM model are used to quantify the contribution of transports in shallow water to the total transport of the Brazil Current in the study region. Due to its pathway (Fig. 3), this contribution will depend on the latitude. An indication of this can be seen in Figure 4, where the agreements are best near the southern latitudes where the confluence with the Malvinas Current results in the separation of the Brazil Current from the shelf break. Based on the grid resolution of $0.5^o$ in Argo & SSH and the slope of the topography, 600 m is used in the following to split the Brazil Current transport into the shallow and open ocean contributions.

For the XBT transects, the analysis was done for two regions separated by the latitude of 27$^o$S. This latitude can be seen as representative for the transition from lower to higher transports. In addition, this latitude is the one where the integration depth transitions from 400 m to 800 m as explained in section 3. In the southern region, the mean contribution of the shallow regions to the Brazil Current transport is 1.7±2.2 Sv (based on 20 transects). Comparing transports of the Brazil Current with and without the shallow region reveals that in 12% of the cases these transports are identical. An additional 44% of the cases have differences that do not exceed 10% of the transport in the Brazil Current. In the northern region, the mean contribution of the shallow regions to the Brazil Current transport is similar with 1.6±1.7 Sv (based on 8 transects). No further analysis is possible in this latitude range because of the small number of transects.

For HYCOM, the focus for quantifying the impact of the transport in shallow regions is on the three latitudes for which the time series are analyzed in detail. At 38$^o$S, the impact of the shallow areas on the transport is negligible (the mean difference is insignificant; identical transports in 86% of the cases), because the Brazil Current is separated from the shelf break most of the time. At 24$^o$S, the impact of the shallow areas is slightly larger (mean difference of 0.7±1.3 Sv; identical transports in 67% of the cases). The largest impact exists at 35$^o$S, where the mean difference is 2.0±1.3 Sv (identical transports in only 14% of the cases). Overall, there is no statistical significant time dependence of the differences. All of these transport reductions are smaller than the differences between the transports from HYCOM and Argo & SSH (Table 1 and Fig. 5).