# Peer review of "Transport Variability of the Brazil Current from Observations and a Model"

_Ocean Science, 2017_

## Referee Comment (RC1) · Anonymous Referee #1 · 19 Sep 2017

The manuscript by Schmid and Majumder uses observational estimates and an ocean model run to evaluate the transport and variability of the Brazil Current (BC). The authors have put together an interesting data set but results are inconclusive. The authors describe the BC in their data, agreeing reasonably well with previous estimates and compare their observational values with an ocean model experiment, which however is poorly described in the manuscript. The interannual variability is also poorly described, with a superficial analysis and correlation with climate indeces. The description of these correlations is either very thin on dynamical explanation or simply not informative. For example, the authors correlate the BC with the AMO, noting that correlations are not robust as expected given that the AMO is an index based on North Atlantic SST anomalies.

[Figure]

General Remarks

The model run is poorly described. How was the model forced? why are the authors using a reanalysis for the period 1993-2012 and then an analysis (free run?) for 2013-2015? A free run and a data assimilating model are two different models, which often have little in common. The model data, as used in the manuscript, do not add any relevant information or mechanism. I suggest to either thoroughly compare observations and model, and use the model data to perform more in-depth analysis, or simply discard model results and focus on the observational data set and its comparison with previous estimates.

The observational data sets, as presented, are confusing. Temperature and Salinity go from 2000 to 2015. Velocities from 1989 to 2016. Satellite data from 1993 to 2015. Most figures are presented for the period 1993-2015 so you should just present that common period.

Minor points

page 2 line 4: The mean transport of the Brazil Current RANGES from 3.8 . . . page 2 line 13: It would be useful to state in the abstract what is the main result of the EOF analysis. page 4 line 10: not sure what you mean here. The Western Boundary Current is not in Sverdrup balance. page 8 line 9: what are all these experiments? no information is given on the model, how it was forced, for how long and whether it is reproducing the BC realistically. page 9 line 8-10: some text is missing here or the English should be improved. page 9 line 12: it seems to me from Fig.3 that the model produces a vigorous BC further north, and that 25S would be more appropriate. page 9 line 16: I think you should clearly state that the model seems to produce feeding westward currents that are located further north, say at 26S and 21S. page 9 line 20: what is the 'full' period? page 9 line 21: Who is 'they'? page 11 line 16: (Figure 4) It is also true that previous estimates seem to agree better with the model and fall within the simulated standard deviation (red) rather than the observational data set

(black). Given that the model has a very fine horizontal resolution, and presumably it is simulating most eddy activity in the region, it could well be that the observational data are giving you too-weak transports. page 12 line 6: TOPEX/POSEIDON page 12 line 10: You have concluded before that the BC is not well defined north of 25S as it is not fed by the westward current. So why discuss here the 'northern region' north of 25S? page 13 line 20: showing the EKE from the obs and the model seems very important so I don't understand why the authors have decided not to show this. page 14 line 7: why have you chosen these three latitudes: 24S, 35S and 38S? page 15 line 5: you give an estimate of 7.9 to 26.2 Sv, but these values are not reproduced in Fig.5, which shows a max value of around 21Sv. If it is due to filtering the authors should clearly state that they are talking about unfiltered data. page 15 line 19-22: sentence very difficult to read. I suggest rewriting this paragraph. page 18 line 1-6: This is very superficial and not useful. Maybe a power spectrum cold give an indication of the variability and its significance? page 18 line 8: why did you choose these indices: SAM, NINO3.4 and AMO? why not local interannual indices like the Atlantic Meridional Mode or the South Atlantic dipole indices? Later, little dynamical explanation is given for the correlation, or lack of, between these indices and the BC. page 19 line 9: the AMO is a North Atlantic index. So why did you use it? what were you expecting to find? I suggest removing any discussion on the AMO. page 19 line 10-12: I don't find this conclusion convincing. You find the largest correlations where the BC is weak and not entirely formed. At 24S the BC has not received most of the westward current flow. You should look at the core of the BC and find significant correlations. page 21 line 12: what is the mechanism proposed by Lopez et al.(2016)?

page 21 line 12: I find it hard to find the 'scope' of this manuscript. What are the main goals and conclusions of this manuscript? they should be stated and clearly presented. page 22 line 3: I would remove the mode data as they do not add anything to the conclusions. page 22 line 22: very vague statement on the variability of the current. page 23 line 8: again, please remove any analysis related with the AMO

---

## Referee Comment (RC2) · Anonymous Referee #2 · 20 Dec 2017

Transport Variability of the Brazil Current from Observations and a Model

Claudia Schmid and Sudip Majumder

General comments The paper addresses the variability of the western boundary current of the South Atlantic Ocean, the Brazil Current. The basic information used is in situ data from Argo profiles and SSH from satellite, as well as results from the HYCOM NCODA system, with 1/12 of degree and a robust assimilation scheme. Moreover, it is also investigated the relationship with climatic indexes such as SAM, Nino3.4 and AMO. The article title sounds very strange with this "a Model", not only because HYCOM NCODA is a well-known and recognized numerical system for ocean circulation, but also considering the bunch of models being used nowadays. My suggestion would be something like ". . . from Observations and an Assimilating System for Ocean Cir-

culation". In my opinion, lots of commas are missing along the text. Moreover, too short phrases are also common, which could be easily merged with the previous one to make the text more fluent and clear. An example of this can be found in lines 21 and 22 of pg19, among others.

Specific comments The methodology for transport estimation as well as for uncertainties quantification has adequate criteria based on previous works of the same group. OK. Lines 5-6, pg14: is there a reason to treat as anomalies the difference between some specific month and the annual value for that year? Is this an anomaly or a seasonal variation? In my opinion, the anomaly should be obtained through the difference between the individual monthly transports and the long term mean for the correspondent month. This aspect needs to be clarified. Maybe an analysis of the anomalies could also bring some interesting aspects of the long term variability, mainly related to the climate indexes and their combination. Lines 22-23, pg 14: a bit forced with "the annual cycle from HYCOM and Argo & SSH are very similar from November until April". Can this be related to the quantification of "anomalies" mentioned in Lines 5-6, pg14? Moreover, there is always a jump between December and January in the mean annual cycles of figure 6. How these figures behave with a long term mean climatology for each month to quantify the anomalies? Line 13, pg15: "In about 2001 to 2010..." should be "Around 2009..." isn't it? Figures 7 and 8: the vectors of the cross wavelet diagrams are impossible to distinguish. Another issue is the absence of the information of wavelet coherence to consider only the some parts of the graphic. This information is crucial and there is a need to present it. Another key issue: is it possible to associate the description of time lags described with the arrow directions in the cross wavelet phase diagrams? This aspect needs to be strongly clarified.

Technical corrections Lines 21-23, pg5: this last phrase should be moved to the end of line 2, pg9, because it is related to methodology. Am I right? Some idea can still remain at the introduction, of course, but not mentioning the appendixes. Line 8 and 15, pg8: two open parenthesis with only one to close; it happens many times, maybe

due to text editor. In any case, it must be corrected.

---

## Author Comment (AC1) · 1 Mar 2018

The manuscript by Schmid and Majumder uses observational estimates and an ocean model run to evaluate the transport and variability of the Brazil Current (BC). The authors have put together an interesting data set but results are inconclusive. The authors describe the BC in their data, agreeing reasonably well with previous estimates and compare their observational values with an ocean model experiment, which how- ever is poorly described in the manuscript. The interannual variability is also poorly described, with a superficial analysis and correlation with climate indeces. The description of these correlations is either very thin on dynamical explanation or simply not informative. For example, the authors correlate the BC with the AMO, noting that correlations are not robust as expected given that the AMO is an index based on North Atlantic SST anomalies.

Response:
Thank you. We expanded the analysis. Detail are provided below
* * *
Please note: all page and line numbers are based on the revised
version in bold font.
* * *
General Remarks 1:
The model run is poorly described. How was the model forced? why are the authors using a reanalysis for the period 1993-2012 and then an analysis (free run?) for 2013-2015? A free run and a data assimilating model are two different models, which often have little in common. The model data, as used in the manuscript, do not add any relevant information or mechanism. I suggest to either thoroughly compare observations and model, and use the model data to perform more in-depth analysis, or simply discard model results and focus on the observational data set and its comparison with previous estimates.

Response:
Thank you. The model was not run by the authors. We revised the text to clarify this and now provide a link to more information (page 8, l.11-19). None of the model runs are free runs (as specified in the manuscript:
"... and uses the Navy Coupled Ocean Data Assimilation (NCODA) system for assimilation." We used the reanalysis version when available. For the more recent years, the reanalysis was not available. Therefore, we used the analysis to get a time series that covers the full time period. No discontinuity is detected at the transition from the reanalysis to the analysis. We expanded the text related to the model throughout the manuscript.
* * *
General Remarks 2:
The observational data sets, as presented, are confusing. Temperature and Salinity go from 2000 to 2015. Velocities from 1989 to 2016. Satellite data from 1993 to 2015. Most figures are presented for the period 1993-2015 so you should just present that common period.

Response:
Thank you, the text has been expanded (page 7, l.6-24; page 8, l.1-4).
The summary is:
We use hydrography and altimetry from 2000-2015 (the period for which we had Argo and SSH data available) to derive the relationship between dynamic height and SSH (SSH ended in 12/2015 at the time we started the computations). We then use the SSH time series 1993-2015 to derive the synthetic dynamic height for the longer time period based on the derived relationship. For the trajectories, we used as much as possible to get a more robust monthly climatology.
* * *
Minor points:
page 2, line 4: The mean transport of the Brazil Current RANGES from 3.8 ...

Response: thank you, revised
* * *
page 2, line 13: It would be useful to state in the abstract what is the main result of the EOF analysis.

Response: thank you, done (page 2, l.13-16)
* * *
page 4 line 10: not sure what you mean here. The Western Boundary Current is not in Sverdrup balance.

Response: Agreed. What we meant is, that the subtropical gyre is largely governed by the Sverdrup Balance. The western boundary current only closes the gyre. We rephrased this (page 4, l.10). What complicates things is that the SEC carries water from the Indian Ocean into the western South Atlantic and bifurcates at the western boundary.
* * *
page 8 line 9: what are all these experiments? no information is given on the model, how it was forced, for how long and whether it is reproducing the BC realistically.

Response: Thank you. These experiments are made available online (the model is not run by us). The paragraph was expanded to clarify this and now points to the web page that makes the experiments available to allow readers to obtain more information (page 8, l.11-19).
* * *
page 9 line 8-10: some text is missing here or the English should be improved.

Response: Thank you – in the process of changing the way the references are shown (transition to the template for Ocean Sciences) the beginning of the sentence starting in the middle of line 9 got lost.
* * *
page 9 line 12: it seems to me from Fig.3 that the model produces a vigorous BC further north, and that 25S would be more appropriate.

Response: thank you, HYCOM shows strengthening of the Brazil Current between 25S and 28S. We revised the text (page 10, l.13-16).
* * *
page 9 line 16: I think you should clearly state that the model seems to produce feeding westward currents that are located further north, say at 26S and 21S.

Response:
Thank you, there indeed is a shift with respect to the latitudes of the westward flow. A sentence was added about this (page 10, l.13-16).
* * *
page 9 line 20: what is the full period?

Response: Thank you. We added the time period at the end of paragraph 4 in section 2 and added "of 23 years" to the end of the sentence (page 10, l.24).
* * *
page 9 line 21: Who is they?

Response: Thank you. "they" = "the layer thicknesses" from the previous sentence. The sentence was revised (page 11, l.1).
* * *
page 11 line 16: (Figure 4) It is also true that previous estimates seem to agree better with the model and fall within the simulated standard deviation (red) rather than the observational data set (black). Given that the model has a very fine horizontal resolution, and presumably it is simulating most eddy activity in the region, it could well be that the observational data are giving you too-weak transports.

Response: Yes, the model is eddy-resolving. However, it seems like the other factors are more important in the comparison of quasi-synoptic surveys (from previous studies) with monthly means (from Argo & SSH and HYCOM).
* * *
page 12 line 6: TOPEX/POSEIDON

Response: Thank you, corrected (page 13, l.10).
* * *
page 12 line 10: You have concluded before that the BC is not well defined north of 25S as it is not fed by the westward current. So why discuss here the "northern region" north of 25S?

Response: Thank you, we rewrote parts of the first paragraph in section 3 (page 10, l.2-11) to make it more clear that the BC is poorly defined in the mean field because of the eddy variability, shifts in its location as well as

Printed by Claudia Schmid

its relative weakness. Even so this is the case, one can estimate the BC
transport and study its variability.
* * *
page 13 line 20: showing the EKE from the obs and the model seems very important
so I don't understand why the authors have decided not to show this.

Response: thank you. We did not show the EKE fields, in the manuscript because
we already had a lot of figures. To show them without making the manuscript
much longer we decided to present EKE figures as a supplement.
* * *
page 14 line 7: why have you chosen these three latitudes: 24S, 35S and 38S?

Response: We picked one latitude far north of the confluence in the regime
dominated by small transports and two in the vicinity of the confluence
(the regime dominated by high transports and high variability).
This information has been added to the manuscript (page 15, l. 8-10).
* * *
page 15 line 5: you give an estimate of 7.9 to 26.2 Sv, but these values are not
reproduced in Fig.5, which shows a max value of around 21Sv. If it is due to
filtering the authors should clearly state that they are talking about
unfiltered data.

Response: Thank you for catching this. The numbers in the table for the
observations at this latitude were wrong. While revising the table we also redid
the computations focused on the interannual variability. In addition to that, we
cross-checked all numbers and made corrections wherever necessary.
* * *
page 15 line 19-22: sentence very difficult to read. I suggest rewriting this
paragraph.

Response: Thank you, we rephrased this (page 17, l.11-16).
* * *
page 18 line 1-6: This is very superficial and not useful. Maybe a power
spectrum cold give an indication of the variability and its significance?

Response:
We expanded the part about the wavelet analysis in the previous section
and rewrote large parts of this section (page 19-22) including the first
paragraph.
* * *
page 18 line 8: why did you choose these indices: SAM, NINO3.4 and AMO? why not
local interannual indices like the Atlantic Meridional Mode or the South
Atlantic dipole indices? Later, little dynamical explanation is given for the
correlation, or lack of, between these indices and the BC.

Response:
We choose SAM because it includes observations close enough to the subtropics
to potentially have an effect on the gyre strength. We choose NINO3.4 because
of studies by others on the teleconnections between the tropical Pacific and the
South Atlantic.

We agree that including AMO does not provide much insight and thus removed it.

Thank you for suggesting to look at the more local indexes. We added SASD
Mode (SST anomalies averaged within 30-40S, 10-30W minus those averaged
within 15-25S, 0-20W) to this study.

We also looked into a potential role of the Atlantic Meridional Mode and
detected no significant correlation. This was not added.

(page 19-22)
* * *
page 19 line 9: the AMO is a North Atlantic index. So why did you use it? what
were you expecting to find? I suggest removing any discussion on the AMO.

Response: Thank you, as stated above, AMO was removed.
* * *
page 19 line 10-12: I don't find this conclusion convincing. You find the

largest correlations where the BC is weak and not entirely formed. At 24S the BC
has not received most of the westward current flow. You should look at the core
of the BC and find significant correlations.

Response:
We revised most of this section and expanded it (page 19-22).
* * *
page 21 line 12: what is the mechanism proposed by Lopez et al.(2016)?

Response:
We revised most of this section and expanded it to explain in more detail what
is going on (page 22-25).
* * *
page 21 line 12: I find it hard to find the 'scope' of this manuscript. What are
the main goals and conclusions of this manuscript? they should be stated and
clearly presented.

Response:
Thank you. We expanded the analysis, revised large parts of sections 4.4
(page 19-22) and 4.5 (page 22-25) and updated the conclusion and abstract.
* * *
page 22 line 3: I would remove the model data as they do not add anything to the
conclusions.

Response:
We kept the model, expanded the use and revised the conclusions and the abstract.
* * *
page 22 line 22: very vague statement on the variability of the current.

Response:
We added a sentence to this paragraph to make it more clear (page 27, l.5-6).
* * *
page 23 line 8: again, please remove any analysis related with the AMO

Response: Done.

---

## Author Comment (AC2) · 1 Mar 2018

Transport Variability of the Brazil Current from Observations and a Model

Claudia Schmid and Sudip Majumder

General comments

The paper addresses the variability of the western boundary current of the South Atlantic Ocean, the Brazil Current. The basic information used is in situ data from Argo profiles and SSH from satellite, as well as results from the HYCOM NCODA system, with 1/12 of degree and a robust assimilation scheme. Moreover, it is also investigated the relationship with climatic indexes such as SAM, Nino3.4 and AMO.

The article title sounds very strange with this "a Model", not only because HYCOM NCODA is a well-known and recognized numerical system for ocean circulation, but also considering the bunch of models being used nowadays. My suggestion would be something like "... from Observations and an Assimilating System for Ocean Circulation".

Response:
Thank you. We changed the title a bit finding a compromise between the suggestion and the initial title.
* * *
Please note: all page and line numbers are based on the revised
version in bold font.
* * *
In my opinion, lots of commas are missing along the text. Moreover, too short phrases are also common, which could be easily merged with the previous one to make the text more fluent and clear. An example of this can be found in lines 21 and 22 of pg19, among others.

Response:
Thank you. During the revisions we tried to make the text more fluent.
* * *
Specific comments

The methodology for transport estimation as well as for uncertainties quantification has adequate criteria based on previous works of the same group. OK.

Lines 5-6, pg14: is there a reason to treat as anomalies the difference between some specific month and the annual value for that year? Is this an anomaly or a seasonal variation? In my opinion, the anomaly should be obtained through the difference between the individual monthly transports and the long term mean for the correspondent month. This aspect needs to be clarified. Maybe an analysis of the anomalies could also bring some interesting aspects of the long term variability, mainly related to the climate indexes and their combination.

Response: Thank you, we expanded the text to clarify what we did.
(page 15, l.8-12)
* * *
Lines 22-23, pg 14: a bit forced with "the annual cycle from HYCOM and Argo & SSH are very similar from November until April". Can this be related to the quantification of "anomalies" mentioned in Lines 5-6, pg14?

Response: thank you, the text has been revised. (page 16, l.15-20)
* * *
Moreover, there is always a jump between December and January in the mean annual cycles of figure 6. How these figures behave with a long term mean climatology for each month to quantify the anomalies?

Response:
All these jumps are quite small. For example, the difference between the December and January value in at 24S is less than 0.5 Sv. This difference is similar to the difference between the transport in April and May for the observations and smaller than the difference between the transport in March and

April for HYCOM, for example. Also, these jumps are smaller than the standard
error at all 3 latitudes. For the mean annual cycle, we get a similar result
with respect to the timing of the seasonal cycle and a bit larger standard
errors because the interannual variability has not been removed.
* * *
Line 13, pg15: "In about 2001 to 2010 ..." should be "Around 2009. . ." isn't
it?

Response: Thank you, we rephrased this. (page 17, l.5-6)
* * *
Figures 7 and 8: the vectors of the cross wavelet diagrams are impossible to
distinguish. Another issue is the absence of the information of wavelet
coherence to consider only the some parts of the graphic. This information is
crucial and there is a need to present it.

Response: Thank you. We improved the discussion of the wavelet analysis. Figures
of the coherence were added and the arrow style was changed.
* * *
Another key issue: is it possible to associate the description of time lags
described with the arrow directions in the cross wavelet phase diagrams? This
aspect needs to be strongly clarified.

Response: Thank you. While improving the discussion we also looked into this
and added our finding to the text.
(page 20, l.15-17)
* * *
Technical corrections

Lines 21-23, pg5: this last phrase should be moved to the end of line 2, pg9,
because it is related to methodology. Am I right? Some idea can still remain at
the introduction, of course, but not mentioning the appendixes.

Response:
Done. (page 5)
* * *
Line 8 and 15, pg8: two open parenthesis with only one to close; it happens many
times, maybe due to text editor. In any case, it must be corrected.

Response: Thank you. This problem was introduced during the transition to the
way citations are done to the standard used by Copernicus. Unfortunately, we
missed some of the problems. We will make sure to fix/avoid such issues in the
revised version.

---

## Author Response (AR2)

Topic Editor Decision: Publish subject to technical corrections (14 May 2018) by David Steve
ns
Comments to the Author:
The authors have done a good job of addressing the concerns of reviewers. Thus I am happy to
 recommend publication of the manuscript. I have some minor technical corrections which will
 hopefully aid clarity.
* * *
Abstract
* * *
Line 11: remove âM-^@M-^\theâM-^@M-^] leaving âM-^@M-^\âM-^@¦ with increasing latitude (both
 in Argo & SSH and HYCOM), mainly due to mesoscale and interannual variability.âM-^@¦âM-^@
M-^]

Done
* * *
Line 12: insert commas âM-^@M-^\âM-^@¦Argo & SSH, as well as HYCOM, reveal interannual varia
bilityâM-^@¦âM-^@M-^]

Done
* * *
Lines 13/4: remove the sentence âM-^@M-^\It mostly does not quite reach the level of signifi
cance because the duration of the multi-year phases with high (low) transports varies quite
a bit.âM-^@M-^] as it is not particularly clear and does not enhance the abstract.

Done
* * *
Line 17: remove âM-^@M-^\(EOF)âM-^@M-^] as the acronym is not used in the abstract and it is
 defined when first used in the main text.

Done
* * *
Page 5, lines 6/7: replace last sentence with âM-^@M-^\Most of the estimates from earlier st
udies are based on quasi-synoptic sections, while some are based on time series from mooring
s with current meters or Inverted Echo Sounders (IES).âM-^@M-^]

Done
* * *
Page 5, line 18. capital N for âM-^@M-^\North AtlanticâM-^@M-^]

Done
* * *
Page 6, line 4: replace âM-^@M-^\dynamicâM-^@M-^] with âM-^@M-^\dynamicsâM-^@M-^] and âM-^@
M-^\isâM-^@M-^] by âM-^@M-^\areâM-^@M-^]

Done
* * *
Page 12, line 14: insert the word âM-^@M-^\magnitudeâM-^@M-^] between âM-^@M-^\withâM-^@M-^]
 and âM-^@M-^\20âM-^@M-^]

Done
* * *
Page 16, line 15: replace âM-^@M-^\in inâM-^@M-^] with âM-^@M-^\inâM-^@M-^]

Done
* * *
Page 20, line 8: SAM isnâM-^@M-^Yt an ocean index. The Marshall time series that is used in
the manuscript is based on land station observations of atmosphere pressure.

We double-checked. Many publications and websites call it an index.
Examples are:

https://instaar.colorado.edu/uploads/publications/lovenduski_grl_2005.pdf
https://journals.ametsoc.org/doi/full/10.1175/2008JCLI2260.1
https://stateoftheocean.osmc.noaa.gov/atm/sam.php
* * *
Page 20, line 10: replace âM-^@M-^\indexesâM-^@M-^] with âM-^@M-^\indicesâM-^@M-^]

Done (throughout the manuscript)
* * *
Page 20, line 21: replace âM-^@M-^\maximumâM-^@M-^] with âM-^@M-^\maximaâM-^@M-^]

Done
* * *
Page 21, line 3: replace âM-^@M-^\indexesâM-^@M-^] with âM-^@M-^\indicesâM-^@M-^]

Done
* * *
Page 26, line 21: replace âM-^@M-^\In consistencyâM-^@M-^] with âM-^@M-^\ConsistentâM-^@M-^]

Done
* * *
Page 33: There is a comma missing after the Evans and Signorini (1985) reference.

Done
* * *
Page 34: change end of b) to âM-^@M-^\âM-^@¦ collected between January 9, 1992 and May 2, 20
16. Though why so precise here when a) just states the years.

2000-2015 stands for January 1, 2000 to December 31, 2015.

We gave the exact dates for the latter, because the time period
deviate from full calendar years.
We changed it to: "collected during January 1992 to April 2016"
* * *
Page 36, caption: Either use grey or gray consistently, better grey. The same holds througho
ut the manuscript.

Done
* * *
Page 39, caption: There is a stray 2 in the word âM-^@M-^\transportâM-^@M-^]

Done
* * *
Page 49, line 3: remove âM-^@M-^\less thanâM-^@M-^]

Done
* * *
Page 49, line 10: extraneous open bracket before the references

Done
* * *
Page 49, line 16: replace âM-^@M-^\BecauseâM-^@M-^] with âM-^@M-^\AsâM-^@M-^]

Done
* * *
Page 49, line 16: replace âM-^@M-^\Because ofâM-^@M-^] with âM-^@M-^\Due toâM-^@M-^]

I assume this refers to line 19. Done
* * *
Page 50, in a few locations. Should you use the phrase âM-^@M-^\identical transportsâM-^@
M-^] presumably the numerical values arising fem the XBT transacts or model are not precisel
y identical, but simply very close.

The transport are identical if none of the transport of
the Brazil Current occurs in the shallow regions.

We rephrased the sentences in the second paragraph to clarify the text
(by defining what we are comparing).
* * *
A general point on the shaded colour figures. Could you avoid using the old default Matlab (
IâM-^@M-^Ym assuming) jet/rainbow colour map. This has a number of issues associated with in
accessibility and perception skewing. Even Matlab have acknowledged this now by making parul
a the default colourmap. See
https://blogs.mathworks.com/steve/2014/10/20/a-new-colormap-for-matlab-part-2-troubles-with-
rainbows/
https://blogs.egu.eu/divisions/gd/2017/08/23/the-rainbow-colour-map/
https://www.climate-lab-book.ac.uk/2014/end-of-the-rainbow/

We tried different colormaps (including red-blue and parula).
The colormap jet works better for representing maxima and minima,
which are the focus of this paper.